# The Role of NEDD4 E3 Ubiquitin–Protein Ligases in Parkinson’s Disease

**DOI:** 10.3390/genes13030513

**Published:** 2022-03-14

**Authors:** James A. Conway, Grant Kinsman, Edgar R. Kramer

**Affiliations:** Faculty of Health, Plymouth Institute of Health and Care Research, Peninsula Medical School, University of Plymouth, Plymouth PL6 8BU, UK; james.conway@plymouth.ac.uk (J.A.C.); grant.kinsman@postgrad.plymouth.ac.uk (G.K.)

**Keywords:** NEDD4, E3 ubiquitin–protein ligase, α-synuclein, Parkinson’s disease, dopaminergic neurons

## Abstract

Parkinson’s disease (PD) is a debilitating neurodegenerative disease that causes a great clinical burden. However, its exact molecular pathologies are not fully understood. Whilst there are a number of avenues for research into slowing, halting, or reversing PD, one central idea is to enhance the clearance of the proposed aetiological protein, oligomeric α-synuclein. Oligomeric α-synuclein is the main constituent protein in Lewy bodies and neurites and is considered neurotoxic. Multiple E3 ubiquitin-protein ligases, including the NEDD4 (neural precursor cell expressed developmentally downregulated protein 4) family, parkin, SIAH (mammalian homologues of *Drosophila* seven in absentia), CHIP (carboxy-terminus of Hsc70 interacting protein), and SCF^FXBL5^ SCF ubiquitin ligase assembled by the S-phase kinase-associated protein (SKP1), cullin-1 (Cul1), a zinc-binding RING finger protein, and the F-box domain/Leucine-rich repeat protein 5-containing protein FBXL5), have been shown to be able to ubiquitinate α-synuclein, influencing its subsequent degradation via the proteasome or lysosome. Here, we explore the link between NEDD4 ligases and PD, which is not only via α-synuclein but further strengthened by several additional substrates and interaction partners. Some members of the NEDD4 family of ligases are thought to crosstalk even with PD-related genes and proteins found to be mutated in familial forms of PD. Mutations in NEDD4 family genes have not been observed in PD patients, most likely because of their essential survival function during development. Following further in vivo studies, it has been thought that NEDD4 ligases may be viable therapeutic targets in PD. NEDD4 family members could clear toxic proteins, enhancing cell survival and slowing disease progression, or might diminish beneficial proteins, reducing cell survival and accelerating disease progression. Here, we review studies to date on the expression and function of NEDD4 ubiquitin ligases in the brain and their possible impact on PD pathology.

## 1. Introduction

Parkinson’s disease (PD) is characterised by the loss of midbrain dopaminergic neurons in the substantia nigra, which is frequently accompanied by an accumulation of α-synuclein in β-sheet filaments in these neurons (so-called Lewy bodies) and neurites [1]. This aggregation process is thought to underlie the disease’s toxicity, with intermediate α-synuclein oligomers being the toxic agent [1]. The accumulation of misfolded α-synuclein in PD is considered to be due to increased expression [2,3] or reduced degradation via the ubiquitin proteasome, the lysosome, and the autophagy system [4,5].

α-synuclein is known to undergo mono- and polyubiquitination; the former modification is normally involved in regulating protein trafficking, and the latter is considered a prerequisite for degradation. However, for small proteins such as α-synuclein, monoubiquitination seems to be sufficient for proteasomal degradation [4,5,6,7] (Figure 1). For α-synuclein, it has been proposed that the non-ubiquitinated protein might be slowly degraded by autophagy, the monoubiquitinated protein might be degraded by the proteasome, and the polyubiquitinated protein may be degraded by the proteasome and lysosome [4,8]. α-synuclein can be ubiquitinated on nine different lysine residues, lysines 6, 10, 12, 21, 23, 32, 34, 43 and 96, with different preferences in monomeric, oligomeric, and aggregated α-synuclein, as N-terminal monoubiquitination stimulates aggregation and proteasomal degradation [4,5].

The attachment of ubiquitin to proteins (“ubiquitination”) is usually catalysed by an enzymatic cascade of a ubiquitin-activating enzyme E1 (only two in the human genome), a ubiquitin-binding/conjugating enzyme E2 (around 35 in the human genome), and a ubiquitin–protein ligase enzyme E3 (around 600 in the human genome) that catalyses the transfer of the C-terminal carboxyl group of ubiquitin to the lysine (K) ε-amino group of the specific substrate. E3 ligases have at least two domains: a region to interact with an E2 enzyme and a region to recognise the specific substrate proteins. Based on the E2 interaction domain, E3 enzymes can be grouped into two families, HECT (homologous to human papillomavirus oncogene E6-associated protein carboxy-terminus) domain E3s and the more frequent single- and multisubunit RING (really interesting new gene; two zinc ions in a cross-braced arrangement of eight cysteines and histidines) and RING-like (U-box found in the polyubiquitin chain elongation protein E4 saccharomyces cerevisiae Ufd2 protein with noncovalent interactions of core amino acids forming a RING-like tertiary structure without zinc and plant homeodomain/proline-hydroxylase-domain/leukaemia-associated protein (PHD/LAP) with zinc) domain E3s [10]. 

Several different E3 ubiquitin–protein ligases have been described to be able to ubiquitinate different forms of α-synuclein [5], but it remains a matter of debate which E3 ubiquitin–protein ligases might be crucial for α-synuclein degradation, how mutations and misfolding of α-synuclein reduce its recognition by E3 enzymes as a substrate, and whether the E3 enzyme activity itself might be altered in PD. Besides the possible redundancy of different E3 ligases, it seems a common theme that E3 ligases ubiquitinate several different substrates. Therefore, accumulated α-synuclein might block E3 ubiquitin ligase activity and lead to the accumulation of other substrates, which may subsequently contribute to the disease aetiology. 

After summarising the different E3 ubiquitin–protein ligases, which have been suggested to use α-synuclein as a substrate, we focus on one group of them—the NEDD4 family, which has many additional PD-linked substrates—and support the idea that NEDD4 family members can be considered as therapeutic targets to treat PD.

## 2. E3 Ligases Ubiquitinating α-Synuclein

Interestingly, members of the single-subunit (parkin, SIAH, CHIP) and multiple-subunit (SCF^FXBL5^) RING domain E3 ligase family, as well as the HECT domain family of E3 ligases (NEDD4 family), have been found capable of ubiquitinating α-synuclein. 

Parkin was the first E3 ubiquitin–protein ligase described to ubiquitinate α-synuclein in vitro and required the presence of the E2 ubiquitin-conjugating enzyme UbcH7 [11]. However, parkin was only able to ubiquitinate a post-translationally modified form of α-synuclein, a specific 22-kilodalton O-glycosylated form of α-synuclein that could also be detected in PD and dementia with Lewy body patients [11]. Parkin was found to be mutated in some familial cases of PD [12], and all parkin mutations seemed to block ubiquitination activity [13]. Parkin is usually autoinhibited, requires self-ubiquitination for its activation and has been shown to label proteins for degradation by the proteasome or lysosome [14]. Parkin has also been shown to ubiquitinate the α-synuclein-interacting protein synphilin-1, which is a presynaptic protein localised to synaptic vesicles, like α-synuclein, and is a constituent of Lewy bodies, like α-synuclein and parkin [15,16]. Recently, α-synuclein was shown to lead to S-nitrosylation, autoubiquitination, and degradation of parkin [17]. However, the relevance of this crosstalk between parkin and α-synuclein to the development and progression of PD remains uncertain.

Next, the two members of the SIAH (mammalian homologues of *Drosophila* seven in absentia) family of E3 ligases, SIAH-1 and SIAH-2 [18], were reported to polyubiquitinate, with the E2 enzyme UbcH5 synphilin-1 promoting their degradation through the ubiquitin–proteasome pathway. α-synuclein was only mono- or diubiquitinated by SIAH-2 and was reported not to be degraded by one laboratory [19] and subsequently degraded via the proteasome pathway by another laboratory [8]. USP9X was shown to be able to remove α-synuclein monoubiquitination generated by SIAH-2 and thereby prevent α-synuclein protein degradation [8]. In addition, SIAH1 was shown to mono- and diubiquitinate α-synuclein on lysines 10, 12, 21, 23, 34, 43, and 96 together with the E2 enzyme UbcH8, which did not affect the degradation of α-synuclein but increased its insolubility, aggregation, and cellular toxicity [20,21]. Interestingly, only the autosomal dominant mutation A30P of α-synuclein in familial PD (and not A53T) abolished SIAH-1 mediated ubiquitination [20]. As for parkin, the in vivo relevance of SIAH-dependent ubiquitination of α-synuclein remains to be shown.

The E3 ligase CHIP (carboxy-terminus of Hsc70 interacting protein) is a multidomain chaperone with a tetratricopeptide/Heat shock protein 70 blinding domain and a U-box/ubiquitin ligase domain [5]. Interestingly, it was shown that the tetratricopeptide repeat domain of CHIP is critical for proteasomal degradation of α-synuclein, whereas the U-box domain of CHIP is sufficient to direct α-synuclein toward the lysosomal degradation pathway [22]. Subsequently, it was suggested that CHIP selectively reduced α-synuclein oligomerisation and toxicity in a tetratricopeptide domain-dependent, U-box-independent manner by specifically degrading toxic α-synuclein oligomers [23]. The ubiquitination of oligomeric α-synuclein by CHIP and UbcH5b can be negatively regulated by the Hsp70-mediated association with the co-chaperone BCL-2-associated athanogene 5 (BAG5) with CHIP [24]. The proof that this is critical for PD pathology still needs to be provided.

More recently, an SCF ubiquitin ligase assembled by the S-phase kinase-associated protein (SKP1), cullin-1 (Cul1), a zinc-binding RING finger protein, and the F-box domain/Leucine-rich repeat protein 5-containing protein FBXL5 (SCF^FXBL5^) was shown to target exogenous α-synuclein and inhibit aggregation in vitro and in vivo in mice [25]. This observation is interesting in regard to alpαha-synuclein seeding and spreading along the gut–brain axis and inside the brain but awaits independent confirmation.

Finally, from the nine human NEDD4 (neural precursor cell expressed developmentally downregulated protein 4) family members, which are NEDD4-1/NEDD4, NEDD4-2/NEDD4L (NEDD4-like), ITCH/AIP4 (itchy/atrophin-1 interacting protein 4), SMURF1 (SMAD-specific E3 ubiquitin–protein ligase 1), SMURF2, WWP1 (WW domain-containing E3 ubiquitin–protein ligase 1), WWP2/AIP2, NEDL1 (NEDD4-like ubiquitin–protein ligase 1), and NEDL2, at least five have been characterised to ubiquitinate α-synuclein and promote its degradation [26,27]. NEDD4-1 together with UbcH5 and UbcH7 used mainly ubiquitin K63 but also K29 and K33 to polyubiquitinate α-synuclein and enhance its lysosomal degradation [28,29] (see Figure 1). Other WW domain/HECT-domain E3s, NEDD4-2, SMURF1, and SMURF2, were reportedly unable to ubiquitinate α-synuclein to the same extent as NEDD4-1 [28]. As detailed below, NEDD4 ligases have three or four tryptophan-rich (WW) domains that mediate protein–protein interactions with an xPxY (PY) motif (often PPxY or LPSY with x being any amino acid) motif on substrates and adaptors. α-synuclein does not contain a PY sequence and instead has proline-rich regions near its C-terminus [27]. It has been proposed that these stretches may mediate recognition of α-synuclein by NEDD4 ligases [28,29]. Recognition of α-synuclein by NEDD4 family enzymes is thought to happen not only through conventional binding to the WW domains of NEDD4 but through the C2 and HECT domains of NEDD4 [27]. It has previously been demonstrated that NEDD4-1 recognises the C-terminus of α-synuclein and subsequently also ubiquitinates α-synuclein on K21 and K96 [28]. Interestingly, overexpression of NEDD4-1 in *Drosophila* rescued the rough eye phenotype induced by α-synuclein overexpression, and in rats, adeno-associated virus (AAV)-mediated NEDD4-1 expression rescued the loss of midbrain dopaminergic neurons induced by AAV-mediated overexpression of human A53T α-synuclein [30] (see also Table 1, Table 2 and Table 3).

More recently, in vitro-generated, β-sheet-containing α-synuclein filaments were found to be a better substrate for ubiquitination than monomeric α-synuclein, and wild-type α-synuclein was observed to be a better substrate than the mutated human A53T α-synuclein when testing the NEDD4 family members NEDD4-1, NEDD4-2, ITCH, SMURF2, and WWP2 [27]. Fibrils of α-synuclein enter the cytosol through a dynamin-dependent mechanism or by penetrating the plasma membrane directly [29]. NEDD4-1 in the cytosol binds the c-terminus of cytosolic α-synuclein through its WW, C2, and HECT domains and preferentially ligates a lysine^63^-linked polyubiquitin chain to the protein. This ubiquitination facilitates the targeting of α-synuclein to endosomes. The ESCRT (endosomal sorting complex required for transport) complex then recognises the ubiquitinated α-synuclein and subsequently transports it to the late endosome via invagination of the endosomal membrane. This may then promote lysosomal degradation of the α-synuclein. The A53T mutation is located close to a region that is thought to form the core of the β-sheet-rich region [183]. It has been suggested that the A53T mutation may reduce the surface hydrophobicity of the β-sheet structure, in turn hindering binding of the ligase [27]. It may be that patients with the A53T mutation develop early-onset PD because mutant A53T α-synuclein forms filaments more rapidly than the wild-type protein [184]. In addition, it is plausible that lack of recognition, ubiquitination, and degradation may contribute to the accumulation and spread of A53T α-synuclein, which the wildtype protein is not subject to (see also Table 1, Table 2 and Table 3).

Interestingly, the Lindquist group developed a phenotypic model of α-synuclein toxicity in yeast. They discovered a small molecule, N-arylbenzimidazole (NAB), that was able to alleviate many major phenotypic markers of α-synuclein toxicity [73,185]. Counter genetic screening showed that NAB activity was dependent on the yeast NEDD4 orthologue Rsp5. A further investigation in mammalian cell models indicated that NAB activity was conserved through evolution and was dependent on NEDD4-1 in these cells. An NAB derivative, NAB2, was found through structure–activity relationship optimisation of the NAB scaffold. NAB2 exhibited improved activity over that of NAB [73,185]. Although NEDD4-1 has a potential role in the cellular response to α-synuclein toxicity, it is considered a noncanonical drug target, since it lacks a discrete active site and has a relatively complex mechanism of activation involving multiple additional enzymes. Despite its complex requirements for activation, NEDD4-1 is thought to be the only member of this signalling pathway that directly interacts with substrates. This allows for great specificity for the manipulation of ubiquitination by drugs. A recent study showed that NAB2 engages with NEDD4-1 at its N-terminus [72]. Treatment with NAB2 significantly increases co-localisation of NEDD4-1 with the early endosome marker Rab5a. This may complement data that have shown that NEDD4-1 traffics α-synuclein to the endosome via K63-linked ubiquitination [28]; however, this should be explored further. α-synuclein toxicity in SH-SY5Y cells was also analysed, and it was found that the trafficking from ER (endoplasmic reticulum) to Golgi regulator (TFG), which is known to regulate ER to Golgi trafficking (a process disrupted in PD), is also an interacting protein of NEDD4-1 [72]. In short, studies into NAB/NAB2–NEDD4-1–α-synuclein interaction show promise for reducing α-synuclein load and toxicity, providing some hope for translation into PD patients in the future (see also Table 1, Table 2 and Table 3).

This summary of different α-synuclein ubiquitinating E3 ubiquitin–protein ligases suggests a possible redundancy in E3 ubiquitin–protein ligases—though they might have different preferences for modified or aggregated forms of α-synuclein—and highlights the importance of validating the significance of these data in vivo under physiological and pathophysiological conditions. Interestingly, so far, in none of the single knockout mice for these E3 ubiquitin–protein ligases has a clear PD phenotype such as α-synuclein accumulation or midbrain dopaminergic cell death, been described, neither in mice deficient for parkin [186], SIAH1a [187], SIAH1b [188], SIAH2 [189], CHIP [190], FBXL5 [191], ITCH [192], SMURF1 [193,194], SMURF2 [194], WWP1 [195], WWP2 [196], NEDD4-1 [197], nor NEDD4-2 [198]. In SIAH1b-, NEDD4-1-, and NEDD4-2-deficient mice, embryonic lethality may have interfered with a careful analysis of the adult dopaminergic system, and conditional knockout approaches for these genes might allow for the investigation of PD-related phenotypes in the near future [197,198,199]. These E3 ubiquitin–protein ligases have been shown to ubiquitinate and thereby regulate not only α-synuclein but other important proteins in the midbrain dopaminergic system, which might also contribute to PD pathology. So far, no mutations or single nucleotide polymorphisms (SNPs) in *Siah1b*, *NEDD4-1*, or *NEDD4-2* have been associated with PD, but these ligases are likely to be important players in the protein network altered in PD [200,201,202]. For SIAH1 function in PD, we refer the reader to the published literature [18,49,52]. Here, we now focus on the two NEDD4 family members NEDD4-1 and NEDD4-2 and discuss their expression, structure, regulation, substrates, and function in the midbrain dopaminergic system, as well as their links to the pathology and treatment of PD.

## 3. NEDD4-1 and NEDD4-2 Expression

The human *NEDD4-1* gene is located on chromosome 15q21.3 and is comprised of 33 exons transcribed in three mRNAs of 6.4, 7.8, and 9.5 kbp in size. It encodes the NEDD4-1 protein, which has a molecular weight of around 120 kDa [203]. In mice, the *NEDD4-1* gene is located on chromosome 9, and the protein has a similar molecular weight [204]. 

The *NEDD4-2* (NEDD4L) gene is located on chromosome 18q12.31 in humans, with 40 exons, and might result in at least five different transcripts, which appear tissue dependent [205,206]. Variability in these transcripts exists in the N-termini, with varying WW domains and sgk1 phosphorylation sites [206,207]. NEDD4-2 has been detected to be marginally smaller than NEDD4-1, with NEDD4-2-specific antibodies detected in two bands on a Western blot. In most tissues, these bands lie at the ~110–115 kDa mark, with one varying in size depending on the tissue type it is expressed in [207,208]. In mice, the NEDD4-2 gene is also localised on chromosome 18. The human *NEDD4-2* gene is around 78% homologue to the human *NEDD4-1* gene, and the proteins NEDD4-1 and NEDD4-2 share 63% sequence identity. *NEDD4-1* gene homologues can be found in all eukaryotic organisms, although *NEDD4-2* is found only in vertebrates. It is therefore thought that *NEDD4-2* arose much later in evolution by gene duplication [42,209].

The two *NEDD4-1* and *NEDD4-2* cDNAs are highly expressed in the developing embryonic and postnatal mouse brain and are subsequently downregulated in the adult brain [210].

NEDD4-1 is also ubiquitously expressed in humans, in the endocrine tissue, lung, proximal digestive tract, gastrointestinal tract, liver, gallbladder, pancreas, kidney, urinary bladder, gonads, muscle, skin, bone marrow, and lymphoid tissue [211]. NEDD4-1 protein was detected in the dopaminergic system in neuromelanin-positive neurons and in reactive glia cells in the substantia nigra and locus coeruleus of Parkinson’s disease and in Lewy body dementia patient brains containing Lewy bodies. It was also detected in lower amounts in neuromelanin-positive neurons in human control brains [28,85]. *NEDD4-1* mRNA was shown to be increased in brain regions with Lewy body pathology [212]. This suggests an important role for NEDD4-1 in disease. It may be a possibility that NEDD4-1 accumulation in Lewy body-containing neurons occurs as a result of neuronal damage. However, a more likely explanation is that NEDD4-1 regulation is representative of a neuroprotective response that leads to a reduction in α-synuclein accumulation. NEDD4-1 mRNA and protein were also detected in the brain stem of mice [197], but so far, no detailed cellular expression study of NEDD4-1 has been conducted in the dopaminergic system of mice. In other parts of the central nervous system NEDD4-1 has been described to be expressed in oligodendrocytes [110]. In cultured cells, NEDD4-1 is predominantly expressed in the cytosol, near the nucleus, and can be found in neurites after neuronal differentiation. It can, however, be recruited with E2 enzymes to the nucleus [203]. NEDD4-1 has also been shown to be active at the cell membrane and exosomes [213]. NEDD4-2 is broadly expressed in humans, in the endocrine tissue, heart, lung, gastrointestinal tract, liver, gallbladder, pancreas, kidney, urinary bladder, gonads, muscle, bone marrow, and lymphoid tissue [211,214]. In the dopaminergic system, NEDD4-2 protein expression was confirmed in the substantia nigra of mice [114] but not in human brain sections. The NEDD4-2 protein was found mainly in the cytosol of substantia nigra neurons and astrocytes [114].

The precise cell type-specific expression of NEDD4-1 and NEDD4-2 in the midbrain dopaminergic system requires further analysis. However, their expression in midbrain dopaminergic neurons not only during development but during adulthood and ageing, as well as in PD brain samples, supports the idea of an important role here for protein homeostasis.

## 4. NEDD4-1 and NEDD4-2 Structure und Posttranslational Modifications

NEDD4-1 and NEDD4-2 proteins have modular structures, which are conserved throughout the family (see Figure 2). These modules consist of a C2 calcium-dependent phospholipid-binding domain at the N-terminus (mediating plasma membrane localisation), which can be involved in targeting substrates and adaptors; three or four WW domains, which mediate protein–protein interactions with an xPxY (variable amino acid-proline-variable amino acid-tyrosine) motif on substrates and adaptors; and a catalytic HECT domain at the C-terminus (catalytic cysteine Cys^867^ in NEDD4-1 and Cys^942^ in NEDD4-2), which forms a thioester bond with activated ubiquitin that has been transferred from an E2 conjugase before transferring ubiquitin moieties to specific substrates [26,42] (see Figure 2). The WW3 and WW4 domains seem to bind to the PY motif in the substrates, with WW3 generally exhibiting higher substrate affinity than WW4 [42]. Recognition of substrates by NEDD4 ligases involves not only the classical E3 ligase binding of PY motifs to WW domains but the C2 and HECT domains of the ligase [27]. The C2 domain can be cleaved off by caspases during apoptosis, allowing fast degradation of the leftover WW and HECT domains [35]. Details on substrates, adaptors, modifiers, and regulators of NEDD4-1 (see Table 1), NEDD4-2 (see Table 2), and both E3 ubiquitin ligases (see Table 3) are summarised in the Table 1, Table 2 and Table 3.

Human NEDD4-1 and NEDD4-2 were shown to work together with the same E2 ligases: Ubc4, UbcH5A, UbcH5B, UbcH5C, UbcH6, and UbcH7 [203,216]. NEDD4-1 can mono- and polyubiquitinate substrates with K48 and K63, but is also involved in K6 and K27 linkage [111]. NEDD4-2 can also mono- and polyubiquitinate its substrate [214].

The activity of NEDD4-1 and NEDD4-2 is normally blocked via autoinhibition, which also stabilises NEDD4 proteins by preventing autoubiquitination and subsequent proteasomal degradation [127,217]. To form this inhibitory protein conformation, the C2 domain can bind the HECT domain in NEDD4-1 [218], or two conserved tryptophans in the WW domain can bind the PY substrate recognition motif in the HECT domain in NEDD4-2 [127] (see also Table 1 and Table 2).

The autoinhibitory conformation can be disrupted, and the substrate specificity altered, by posttranslational modifications including phosphorylation, ubiquitination, neddylation, and SUMOylating, as well as by the presence of calcium binding to the C2 domain of NEDD4-1 [219], autoubiquitination of NEDD4-2 [127], or binding of adaptor and scaffold proteins such as 14-3-3, Numb, or NEDD4 family-interacting proteins (NDFIP1 and NDFIP2) [93,220,221]. In general, substrate-binding seems able to change the conformation of NEDD4 proteins, allowing self-ubiquitination and subsequent degradation, which results in downregulation of NEDD4 once it has ubiquitinated its targets [127].

Autoubiquitination of NEDD4 proteins can be triggered by different interaction partners and can lead to activation, inactivation, or different substrate specificity. The low-density lipoprotein receptor class A domain containing 3 (LRAD3), a member of the low-density lipoprotein (LDL) receptor family, has been found to bind NEDD4 proteins, leading to NEDD4 autoubiquitination and subsequent degradation [222]. Self-catalysed monoubiquitination of NEDD4-1 can enhance substrate recruitment, as shown for the clathrin-coated pit adaptor protein EPS15 (epidermal growth factor receptor substrate 15), which is monoubiquitinated by NEDD4-1 as well as parkin [217,223]. Monoubiquitination of EPS15 leads to an intramolecular binding of ubiquitin to the ubiquitin interaction motive of EPS15 and prevents, for example, EPS15-dependent recruitment of monoubiquitinated EGFR to clathrin-coated pits for internalisation and deactivation [223]. However, ubiquitination of a conserved lysine residue in the HECT domain α1-helix of one NEDD4-1 protein was also suggested to expose the α1-helix to bind to the HECT ubiquitin-binding patch of another NEDD4-1 protein, allowing NEDD4-1 to form an inactive trimer [224]. The binding of the adaptor protein Numb to NEDD4-1 has also been shown to stimulate NEDD4-1-mediated ubiquitination of the tumour suppressor PTEN (phosphatase and tensin homologue) and its subsequent degradation [225].

NEDD4-2 is a target for NEDDylation, which is a similar posttranslational modification process to ubiquitination that conjugates a ubiquitin-like protein, NEDD8 (neural precursor cell-expressed developmentally downregulated gene 8), to a substrate with the help of E1, E2, and E3 enzymes (mainly cullin-RING ubiquitin ligases) [129]. Neddylation of NEDD4-2 increases its ubiquitination activity regarding the sodium-coupled bicarbonate cotransporter 1 (NBCe1), which is essential for acid–base homeostasis in the kidney, and leads to proteasomal degradation of NBCe1 and its translocation from the cell membrane into the cytosol [226].

Preliminary data have suggested that NEDD4-1 can be SUMOylated, which is, again, a similar posttranslational modification process to ubiquitination that links SUMO to a substrate in the presence of E1, E2 (Ubc9), and E3 (Smt3p) enzymes [62]. Sumoylation of NEDD4-1 seems to occur not on the consensus site (K357) but on an unknown site; it decreases NEDD4-1 autoubiquitination activity [62].

Alterations in the phosphorylation of NEDD4-1 and NEDD4-2 have been widely observed to regulate their ubiquitination activity and alter the binding of adaptor or scaffold proteins. Fibroblast growth factor receptor 1 (FGFR1) is a substrate of NEDD4-1 ubiquitination that triggers c-Src kinase-dependent phosphorylation of NEDD4-1 at Tyr^43^ in the C2 domain and Tyr^585^ in the HECT domain, supporting activation [43]. For NEDD4-2, G-protein-coupled receptor (GPCR) protease-activated receptor-1 (PAR1) stimulates c-Src-mediated Tyr^485^ phosphorylation within the 2,3-linker peptide between WW domains 2 and 3 and leads to NEDD4-2 activation [227]. Phosphorylation of *Xenopus* NEDD4-2 on Ser^338^ or Ser^444^ by the serine/threonine kinase serum- and glucocorticoid-induced kinase 1 (SGK1) was shown to lead to a reduction in its affinity for the natural NEDD4 substrate epithelia Na^+^ channel (ENaC), which regulates whole-body Na^+^ balance and blood pressure [228,229]. Human NEDD4-2 phosphorylation by aldosterone-induced SGK1 on Ser^342^ and Ser^448^ (and Thr^367^) was shown to facilitate 14-3-3 protein binding to NEDD4-2, leading to inhibition of the interaction between NEDD4-2’s HECT and WW domains, stabilisation of ENaC in the kidney, and enhanced ubiquitination of the AMPA (α-amino-3-hydroxy-5-methyl-4-isoxazolepropionic acid) receptor subunit GluA1 (glutamate ionotropic receptor AMPA type subunit 1) in the brain. For *Xenopus* NEDD4-2, it was suggested that the 14-3-3 dimer binds first on NEDD4-2 P-Ser^444^, the high-affinity (major) site, and subsequently on one of the lower-affinity (minor) sites, P-Ser^338^ or P-Thr^363^ [230]. For human NEDD4-2 it has been shown that the 14-3-3 dimer simultaneously anchors on two of the three phosphorylation sites, P-Ser^342^, P-Thr^367^ and P-Ser^448^, of NEDD4-2, with P-Ser^448^ being the key residue [93]. SGK1 also leads to phosphorylation of human NEDD4-2 Ser^468^ and an increase in ENaC protein [92,231]. Interestingly, SGK1 has also been suggested to be a NEDD4-2 substrate, leading to its own degradation and generating a negative feedback loop [232]. The same three SGK1 phosphorylation sites, Ser^342^, Ser^448^ and Thr^267^, of NEDD4-2 are also used by vasopressin-induced cyclic AMP-dependent protein kinase A (PKA). Furthermore, insulin activates SGK1 and Akt (protein kinase B) and leads to Ser^342^ and Ser^428^ phosphorylation of human NEDD4-2, upregulating ENaC on the membrane [155]. IkappaB kinase β (IKKβ) has been found not only to bind to ENaC and enhance its activity but to phosphorylate *Xenopus* NEDD4-2 on Ser^444^, preventing NEDD4-2-dependent ENaC ubiquitination [117]. Interestingly, 14-3-3η has also been shown to bind and inhibit the ubiquitination activity of wildtype parkin but not of parkin with R42P, K161N, and T240R mutations associated with autosomal recessive juvenile parkinsonism [233]. The parkin/14-3-3 inhibitory complex could be prevented by wildtype α-synuclein but not by A30P and A53T mutations, causing PD [233]. These data define the chaperone-like protein 14-3-3 as an important inhibitor of E3 ligases associated with PD.

The regulatory mechanisms occurring in the midbrain dopaminergic system to change the activity or substrate specificity of NEDD4-1 and NEDD4-2 have so far not been investigated but are likely to utilise at least some of the aforementioned posttranslational modifications and interaction partners. With further information about important in vivo substrates and functions of NEDD4-1 and NEDD4-2 in dopaminergic neurons, it will be a revealing task to study the detailed regulatory mechanisms.

## 5. NEDD4-1 and NEDD4-2 Substrates, Adaptors, Regulators, and Function

Upon the discovery of NEDD4-2, it was proposed that NEDD4-1 and NEDD4-2 may have redundant functions with shared interaction partners and substrates, however there appears to be adaptors, substrates, and functions specific or unique to NEDD4-1 and NEDD4-2. Below, we list some possible shared and unique NEDD4-1 and NEDD4-2 substrates, adaptors, and regulators.

The different phenotypes of NEDD4-1 and NEDD4-2 knockout (KO) mice suggest that their main substrates are distinct, and the redundancy might be limited to a few substrates and functions [42]. The predominant phenotype of NEDD4-1 KO mice is embryonic lethality at midgestation, with pronounced heart defects (double-outlet right ventricle and atrioventricular cushion defects) and vasculature abnormalities leading to growth retardation (with a body weight less than 40% of that of wild-type littermates) [46,117,118]. In contrast, NEDD4-2 KO mice show perinatal lethality, with increased ENaC levels that seem to cause premature foetal lung fluid clearance, resulting in a failure to inflate the lungs [198]. Only a few of these mice survived up to 22 days [198]. This phenotype was also confirmed in lung-specific NEDD4-2 deficient mice [234]. When crossing floxed NEDD4-2 mice with EIIa-Cre mice (B6.FVB-Tg(EIIa-cre)C5379Lmgd/J) [235] expressing Cre in a mosaic pattern in the embryo before implantation in the uterine wall, the NEDD4-2 KO mice might not be a complete null for NEDD4-2 [236]. These mice were viable but showed defects in the respiratory, renal, cardiac, neural, and immune systems and high blood pressure, indicating that NEDD4-2 is a key regulator of Na^+^ homeostasis and that ENaC is one of its most important physiological substrates [236]. This suggests that even ENaC can be a substrate of NEDD4-1 and NEDD4-2 [42]. The NEDD4-1 and NEDD4-2 KO mouse data suggested that in vivo NEDD4-2 is most likely the more important E3 ubiquitin ligase for ENaC. Interestingly, the G protein-coupled receptor kinase 2 (Grk2) can phosphorylate ENaC on Ser^633^ in the C-terminus of the β-subunit, which increases ENaC activity and prevents ENaC ubiquitination by NEDD4 ligases and subsequent degradation [237]. GRK2 and other GRK family members have also been described to phosphorylate α-synuclein on Serine^129^, which is common in PD patients, however it has not been reported whether this phosphorylation negatively influences ubiquitination by NEDD4 ligases [238].

This suggests that NEDD4-1 and NEDD4-2 may have common but also unique functions, and this might depend on the specific tissue investigated. Therefore, it seems important to investigate how far NEDD4-1 and NEDD4-2 have redundant and unique functions in vivo during the development and maintenance of the midbrain dopaminergic system as well as in pathophysiological conditions leading to PD.

Most NEDD4-1 and NEDD4-2 substrates and adaptors have so far been investigated only in vitro and await in vivo confirmation. PTEN (phosphatase and tension homologue) is a good example to illustrate the importance of verifying possible NEDD4 substrates in vivo under physiological conditions in an organism such as the mouse. Cell culture experiments and human cancer tissue suggested that PTEN might be a NEDD4-1 and NEDD4-2 substrate [177,178,179,180,239]), but analysis of NEDD4-1 and NEDD4-2 knockout mice has shown that PTEN stability, subcellular localisation, and activity are not altered in the absence of NEDD4-1 and/or NEDD4-2 [181,182]. Furthermore, more recent cell culture experiments have not supported PTEN as a NEDD4 substrate [225]. Therefore, more research would be required to finally resolve this controversy between in vitro and in vivo data.

The adaptor proteins NDFIP1 (NEDD4 family-interacting protein 1 or NEDD4 WW domain-binding protein 5 (N4WBP5)) and NDFIP2 (NEDD4 family-interacting protein 2 or NEDD4 WW domain-binding protein 5A (N4WBP5A)) are small, endosomal, PY-motif-containing membrane proteins that can both function as adaptors for NEDD4-1, NEDD4-2, ITCH/AIP4, WWP1, and WWP2, facilitating their binding to proteins that lack PY motifs, preventing autoinhibition of the ligase, and possibly serving as ubiquitination substrates. Overexpression of NDFIP1 is able to recruit NEDD4-1, NEDD4-2, and ITCH to neuronal exosomes, which are normally free of these E3 ligases, for secretion [109,203]. The positive effect of NDFIP1/NEDD4-1 in improving neuronal survival during brain injury suggests that perhaps exosomal NEDD4-1 might enhance transport and degradation of unwanted proteins [203]. As microglial exosomes facilitate the transmission of α-synuclein in PD [240], it would be of interest to study the role of NEDD4 ligases in this process. NDFIP1 and NDFIP2 are physically and functionally associated with multiple components of the epidermal growth factor (EGF) signalling cascade, and their levels modulate the relative output of different signalling pathways. They associate with the EGF receptor and the phosphatase and tension homologue (PTEN) and control the ubiquitination and abundance of PTEN, cellular Casitas B-linage Lymphoma E3 ligase (c-CBL), and cellular Sarcoma family kinases (c-Src). NDFIP2, but not NDFIP1, also binds to and is phosphorylated by two c-Src kinases (Src and Lyn) and can act as a scaffold for Src phosphorylation of NDFIP1 and potentially other substrates. Depletion of NDFIP1 inhibits serine/threonine kinase Akt (protein kinase B, PKB) activation in EGF-stimulated HeLa cells, stimulates activation of cellular transcription factor c-Jun-N-terminal Kinase (Jnk), and enhances cell multiplication. Interestingly, increased iron is often found in the substantia nigra of PD patients and has been associated with increased NDFIP1 levels [239]. It would be of interest to examine whether iron misregulation may serve to be protective to nigral dopaminergic neurons by upregulating NDFIP1 and facilitating NEDD4-1-mediated ubiquitination of α-synuclein.

Adaptor proteins such as NDFIP1 and NDFIP2 seem to use different members of the NEDD4 family in vivo. NDFIP1-deficient mice showed a reduced life expectancy, with severe inflammation of the skin and lung, enhanced T-cell activation, proliferation and differentiation to T helper 2 cells, and a prolonged JunB half-life such as that in in Itchy mutant mice lacking functional ITCH protein [192,241]. NDFIP2-deficient mice showed no overt immunopathology, but NDFIP2 deficiency seemed to enhance the NDFIP1 knockout phenotype, leading to further accumulation of effector CD4+ T cells and an increase in JAK (Janus kinase) protein, which might be explained by reduced Itch or NEDD4-2 activation [242]. Further research has to be done to confirm the use of the adaptor proteins NDFIP1 and NDFIP2 by NEDD4-1 and NEDD4-2.

NEDD4-1 activation has also been shown to be important for autophagy and mitophagy [66,243]. LC3 (MAP1LC3, microtubule-associated protein 1 light chain 3) is essential in autophagy by functioning in elongation of the phagophore double-layer membrane and in the recruitment of proteins for autophagic processes. LC3 activates and recruits NEDD4-1 to the phagophore assembly site (PAS) by binding the conserved WXXL LC3-binding motive between the C2 and the WW2 domains. LC3-I is activated to LC3-II by cleavage and conjugation to phosphatidylethanolamine (PE) and is recruited to autophagosomes by binding LIR (LC-3 interacting region domain-containing protein). Subsequently, NEDD4-1 ubiquitinates the LC3-interacting protein p62 (sequestosome-1, SQSTM1) and beclin-1 (BECN1), which seems required to recruit downstream effectors for autophagosome formation [32,66]. More recently, NEDD4-1 lysine^29^-linked autoubiquitination on lysine^1279^ was shown to recruit USP13 (ubiquitin-specific protease 13) to form a deubiquitination complex, which stabilised VPS34 to promote autophagy by removing the lysine^48^-linked polyubiquitin chains from VPS34 at lysine^419^ [74]. Surprisingly, in mice, endoplasmic reticulum (ER) stress and activation of the unfolded protein response (UPR) increased autophagy and NEDD4-2 expression in the liver, but not NEDD4-1 expression. In addition, in cell culture, high amounts of NEDD4-2 correlated with increased autophagy, while low amounts of NEDD4-2 correlated with reduced autophagy [244]. In PD, reduced autophagy is a common phenotype that can be triggered by α-synuclein accumulation and might be enhanced by NEDD4-1 and NEDD4-2 [245].

Interestingly, NEDD4 ubiquitination activity is required for the release of some retroviruses but might be inhibited as a cellular defence mechanism. The Gag protein of human oncoretrovirus HTLV-1 (human T-lymphotropic virus type 1) has a tandem PPPY/PTAP motif and needs to be ubiquitinated by the E3 ligase NEDD4-1 at the plasma membrane. It also requires Tsg101 (tumour susceptibility gene 101) recruitment at the ESCRT (endosomal sorting complexes required for transport) pathway in late endosomes/multivesicular bodies for driving virus budding [49]. Despite the HIV (human immunodeficiency virus) Gag protein lacking a PY motif, it also uses NEDD4-1 and NEDD4-2 for its ubiquitination, to stimulate budding; Nedd4-2, with the adaptor protein AMOT-1 (angiomotin-1 protein); and NEDD4-1, by binding and ubiquitinating adaptor protein ALIX (apoptosis-linked gene 2 (ALG-2)-interacting protein X, programmed cell death 6-interacting protein) [246]. However, NEDD4 family members might be inhibited in cells after viral or bacterial infection by binding with upregulated interferon-induced ubiquitin-like protein ISG15 (interferon-stimulated gene 15). The binding of ISG15 to NEDD4-1, NEDL1, NEDL2, or WWP2 can block their interaction with ubiquitin-E2 enzymes and interfere with the ubiquitination of retroviral group-specific antigen precursors and matrix proteins, such as VP40 of Ebola with a PPxY motif, which is essential for the release/budding of Ebola, vesicular stomatitis, and rabies virus particles [247,248]. DNA virus proteins such as the latent membrane protein 2A (LMP2A) of Epstein–Barr Virus (EBV, human herpesvirus 4) also interact via their PPPPY motif with the NEDD4 family members NEDD4-1, ITCH, and WWP2 [68]. This interaction leads to the ubiquitination of LMP2A and LMP2A-associated proteins such as the protein tyrosine kinases Lyn (Lck/Yes novel tyrosine kinase of the Src kinase family) and Syk (spleen tyrosine kinase), which might be important for EBV latency and the regulation of B-cell signal transduction [68]. Taken together, this suggests an important role of NEDD4 family members in vesicular transport and retrovirus propagation being regulated by the immune system, which have been found to be altered also in neurodegenerative diseases such as PD [249].

In the context of PD, most research in the past focused on NEDD4-1 after NEDD4-1 was found to ubiquitinate α-synuclein as detailed above [28,29]. An interesting NEDD4-1 substrate in this regard is the receptor tyrosine (and serine) kinase RET (abbreviation for rearranged during transfection), the canonical receptor for the TGF-β (transforming growth factor-β)-related neurotrophic factor family member GDNF (glial cell line-derived neurotrophic factor), which is currently in clinical trials in PD patients [245,250,251,252] (see Figure 1). The RET receptor is important for the maintenance, protection, and regeneration of midbrain dopaminergic neurons [253,254,255]. In cell culture experiments, the turnover of the long splice isoform of RET, RET51, is mediated after activation and autophosphorylation by binding of the adaptor protein GRB2 (growth factor receptor-bound protein 2) to RET tyrosine^1096^ and subsequent recruitment of the E3 ubiquitin–protein ligase c-CBL [82]. However, the short RET isoform, RET9, binds to the adaptor proteins GRB10 or SHANK2 (SH3 And Multiple Ankyrin Repeat Domains 2). This depends on its phosphorylated tyrosine^1062^ and the C-terminal PDZ-binding motif (PDZ is an initialism combining the first letters of the first three proteins discovered to share the domain—postsynaptic density protein (PSD95), Drosophila disc large tumour suppressor (Dlg1), and zonula occludens-1 protein (zo-1)) and recruits NEDD4-1 [82]. RET polyubiquitination triggers receptor internalisation from clathrin-coated pits at the cell membrane into endosomal compartments for receptor recycling to the cell surface or lysosomal degradation [256]. RET51 shows more K63 ubiquitin linkage—in contrast to RET9—and can be sorted to RAB11-positive recycling endosomes for signalling, intracellular trafficking, and return to the cell surface or targeted for lysosomal degradation [257,258]. RET9 ubiquitination chains are more K48 linked, targeting the protein more for proteasomal degradation [259]. Determining whether NEDD4-1-dependent ubiquitination of RET9 also occurs in dopaminergic neurons in vivo and influences survival and physiology requires further investigation.

Other receptor tyrosine kinases have also been suggested to be substrates of NEDD4-1, such as the fibroblast growth factor receptor 1 (FGFR1) [43] and the epidermal growth factor receptor (EGFR) members ErbB1 (erythroblastic leukaemia viral oncogene homologous-B2 receptor tyrosine kinase 1) [225] and HER3/ErbB3 (human epidermal growth factor receptor 3) [53]. NEDD4-1 also mediates the adaptor protein β-arrestin2’s agonist-dependent ubiquitination and lysosomal degradation of the β2-adrenergic receptor (β2AR) [33]. IGF1R (insulin-like growth factor I receptor) can bind the adaptor protein GRB10, and this was suggested to mark IGF1R for NEDD4-1-dependent polyubiquitination and degradation [168,260,261] or protect IGF1R from NEDD4-1 ubiquitination [262]. Further work is needed to understand this controversial NEDD4-1 and IGF1R crosstalk. VEGFR-2 (vascular endothelial growth factors receptor 2), but not VEGFR-1, has also been suggested to be protected from NEDD4-1-induced degradation by binding GRB10, although VEGFR-2 might not be a direct NEDD4-1 ubiquitination substrate [91]. VEGF stimulation of VEGFR-2 increases GRB10 expression and c-Src-dependent tyrosine phosphorylation of GRB10, which subsequently increases VEGFR-2 protein levels [263]. Interestingly, in NEDD4-1 knockout mice, GRB10 protein levels were increased and IGF1 and insulin signalling were reduced, while *GRB10* gene deletion rescued the NEDD4-1 knockout lethality, suggesting a negative regulatory function of GRB10 for IGF1 and insulin signalling [262]. These data suggest that NEDD4-1 might not only directly targets receptor tyrosine kinases as substrates but also indirectly regulate receptor tyrosine kinase signalling by targeting associated adaptor proteins. For example, NEDD4-1 monoubiquitinates insulin receptor substrate (IRS)-2, which promotes its binding to the clathrin-coated pit adaptor protein epsin-1 and the recruitment of IGF1R, which phosphorylates IRS-2, stimulating downstream signalling [59]. Another NEDD4-1 monoubiquitination substrate seems to be the adaptor protein HGS (hepatocyte growth factor-related kinase substrate, HRS), which leads to intramolecular binding of ubiquitin to the HGS ubiquitin-interaction domain (UIM), leading in turn to reduced endocytosis of EGFR [55]. The secretory carrier membrane protein-3 (SCAMP3) has a PY motif and seems to be multimonoubiquitinated by NEDD4-1, which allows HGS interaction and prevents EGFR degradation [87]. The endosomal sorting complexes (ESCRT-0, -I, -II, -III, VPS4-VTA1 (vacuolar protein sorting protein 4 and vesicle trafficking protein 1), and ALIX (apoptosis-linked gene 2-interacting protein X) homodimer) are peripheral membrane protein complexes required together for degradation of damaged or unwanted plasma membrane and cytosolic proteins, lysosome and multivesicular body (MVB) biogenesis, autophagy, and viral budding [264,265] The ESCRT-0 complex sorts ubiquitinated membrane proteins into MVB. It consists of HGS and STAM (signal transduction adaptor molecules STAM1 and STAM2) proteins and can be associated with the ubiquitin-binding domain-containing protein EPS15B (epidermal growth factor receptor pathway substrate 15B) to mediate EGFR degradation [265,266]. EPS15 is associated with clathrin-coated pit adaptor protein 2 (AP2) and plays a role in EGFR internalisation [266]. The NEDD4 family members NEDD4-1 and ITCH/AIP4 have been found associated with ESCRT complexes, which seems important not only for viral GAG protein ubiquitination and budding but for degradation of a membrane-associated pool of Lys^63^ polyubiquitinated α-synuclein [28,265]. ESCRT proteins (VPS4 from ESCRT-0, charge multivesicular body protein 2B (CHMP2B) from ESCRT-III) have been found to be important for lysosomal targeting of α-synuclein and are also localised to Lewy bodies [29,267,268,269]

Both NEDD4-1 and NEDD4-2 can bind the PY motif containing non-receptor tyrosine and serine/threonine kinase ACK (activated Cdc42-associated tyrosine kinase, TNK2). NEDD4-1 leads to polyubiquitination and subsequent lysosomal degradation of ACK along with EGFR in response to EGF stimulation, while the data for NEDD4-2 in regard to ACK ubiquitination remain inconsistent [151,152]. Interestingly, FGFR3 activation leads to NEDD4-1 phosphorylation, which subsequently targets the transmembrane protein programmed death-ligand 1 (PD-L1, cluster of differentiation 274 (CD274), B7 homologue 1 (B7-H1)), which is involved in immune system suppression for Lys^48^-linked polyubiquitination and degradation [270]. Furthermore, activation of the small GTPase RAS (rat sarcoma virus protein) is reduced by NEDD4-1 ubiquitinating the PY motif containing the RAS activator CNrasGEF (cyclic nucleotide RAS guanine-nucleotide exchange factor) for proteasomal degradation [40].

A receptor tyrosine kinase that is suggested to be a NEDD4-2 substrate is the NGF (nerve growth factor) receptor TRKA (tropomyosin receptor kinase A; NTRK1, neurotrophic receptor tyrosine kinase 1) with a PPXY motif, which is, in its activated/phosphorylated state, marked for degradation by NEDD4-2-dependent ubiquitination and is more abundant in the dorsal root ganglia of NEDD4-2 deficient mice [145,271,272]. However, the closely related BDNF (brain-derived neurotrophic receptor) receptor TRKB lacks a PPXY motif and seems not to be a NEDD4-2 substrate [145]. TRKA is normally not expressed in midbrain dopaminergic neurons, but the ectopic expression of TRKA in vivo in mice combined with NGF treatment protected dopaminergic neurons from 6-OHDA-induced cell death [273]. TRKB is found in midbrain dopaminergic neurons but is not essential for development and maintenance [253]. However, TRKB seems to protect cells from MPTP (1-methyl-4-phenyl-1,2,3,6-tetrahydropyridine/1-methyl-4-phenylpyridinium) toxicity in mice [274].

NEDD4-2 and other NEDD4 family members have also been found to regulate the signalling of the transforming growth factor beta (TGF-β) family of ligands (33 human genes encoding, for example, TGF-β 1, 2, and 3; activins; inhibins; and bone morphogenetic protein (BMPs)). TGF-β family members activate the TGF-β receptor by inducing the heterotetramerisation of the two single transmembrane serine–threonine (and tyrosine) kinase receptors, TGF-β receptors type I and type II [275,276]. TGF-β receptor signalling leads to transcriptional regulation of target genes through the canonical SMAD signalling pathway (the SMAD abbreviation refers to the homologies to the *Caenorhabditis elegans* SMA (“small” worm phenotype) and the MAD family (“Mothers Against Decapentaplegic”) of genes in *Drosophila*) protein involving signalling pathway and the non-canonical (SMAD independent, involving tyrosine-autophosphorylation) [275,276]. TGF-β receptor type I has also been described as a NEDD4-2, SMURF1, SMURF2, and WWP1 substrate in conjunction with the inhibitory SMAD7 protein as an adaptor. It is marked for degradation by ubiquitination. The TGF-β I and II receptors signal together through the receptor-regulated SMADs (R-SMADs 1, 2, 3, 5, and 8/9), which can partner with SMAD4 (Co-SMAD) or be suppressed by the inhibitory SMADs (I-SMAD 6 and 7) or the SMAD corepressor SnoN (Ski-relate novel protein N) to downregulate TGF-β receptor signalling [140,276]. Interestingly NEDD4-2 was able to bind to all SMAD proteins with a PY motif (1,2,3,5,6,7), but not to those without one (4 and 8), and it induced ubiquitination and subsequent degradation of SMAD2 but not of SMAD3 [140]. Although NEDD4-2 and SMURF2 can both bind to SnoN via SMAD2, 3, or 4, only SMURF2 can ubiquitinate SnoN and lead to its degradation [140]. The TGF-β I and II receptors are expressed in dopaminergic neurons, and TGF-β II-deficient mice showed reduced TGF-β I receptor levels, reduced dendritic growth and spine formation, a decreased range of excitatory-to-inhibitory synapses, a reduced excitation/inhibitory ratio (ratio of evoked miniature excitatory postsynaptic currents (mEPSC) to miniature inhibitory postsynaptic currents ratio (MIPSC)), hyperactivity, and a reversal-learning defect but no change in dopaminergic cell counts [277].

As for α-synuclein, several different ubiquitin–protein ligases have been proposed to influence internalisation, signalling and degradation of most receptor and cytosolic tyrosine and serine–threonine kinases. Therefore, very careful in vivo analysis is needed to understand the roles of NEDD4-1 and NEDD4-2 in regulating a specific kinase, considering in detail the context, such as the organism, developmental stage and age, tissue, and environmental and physiological challenges.

Concerning PD and the function of the dopaminergic system, it is also important to mention that NEDD4-2 can ubiquitinate neurotransmitter transporters. The dopamine transporter (DAT) is required for the reuptake of dopamine into dopaminergic neurons and has been suggested to be ubiquitinated by NEDD4-2, but not NEDD4-1, leading to endocytosis by binding to epsin and Eps15 on clathrin-coated pits [108,278]. NEDD4-2 seems to cooperate with the E2 enzymes UBE2D and UBE2L3 to conjugate, in a PKC-dependent reaction, primarily lysine^63^-linked ubiquitin chains onto DAT [108]. Reduced DAT levels increase the amount of extracellular dopamine and prolong the stimulation of pre- and postsynaptic dopamine receptors, leading in mice to locomotion defects [279]. Another NEDD4-2 substrate that is ubiquitinated in an PKC-dependent manner and reduced in neurodegenerative diseases is the glutamate transporter-1 (GLT-1), which is subsequently internalised and degraded [280]. In MPP^+^-treated astrocytes and MPTP-treated mice, NEDD4-2 mediated the ubiquitination of GLT-1 [114]. Conversely, NEDD4-2 knockdown increased glutamate transporter protein levels. In MPTP-treated mice, NEDD4-2 knockdown ameliorated movement disorders, increased tyrosine hydroxylase expression in the midbrain, and attenuated astrogliosis and reactive microgliosis associated with glutamate excitotoxicity [114]. To support the idea that NEDD4-2 might be a therapeutic target for the treatment of PD, it would be of interest to confirm these results in conditional NEDD4-2 KO mice treated with MPTP or a more physiological challenge such as mild overexpressed α-synuclein [245].

Both NEDD4-1 and NEDD4-2 seem to ubiquitinate, in a PKC-dependent reaction (phosphorylation of NEDD4-2 threonine^197^, serine^221^, serine^354^, and serine^420^), the human organic anion transporter 1 (hOAT1) in kidney proximal tubule cells, which is important for the release of anti-HIV drugs, anti-tumour drugs, antibiotics, and anti-inflammatory drugs. Ubiquitination of hOAT1 leads to its reduced activity, internalisation and degradation [131,281]. NEDD4-2 has been suggested to also ubiquitinate hOAT3 [282]. NEDD4-1 seems capable of ubiquitinating the ATP binding cassette transporter B1 (ABCB1), which can export the neurotoxic peptide β-amyloid from endothelial cells in the blood–brain barrier to protect the brain [31]. In Alzheimer’s patients, ABCB1 protein levels were reduced, while NEDD4-1 protein levels were increased, suggesting NEDD4-1 as a therapeutic target for the treatment of Alzheimer’s disease.

Another interesting NEDD4-1 substrate is the proapoptotic protein RTP801 (regulated in development and DNA damage responses, Redd1; DNA-damage-inducible transcript 4, DDIT4; or dexamethasone-induced gene 2 encoded protein, Dig2), a mTOR suppressor that has previously been shown to cause neuronal death in both cellular and animal models of PD [85]. RTP801 has been found upregulated in toxin-induced PD animal models such as 6-hydroxydopamine (6-OHDA), MPTP/MPP+ and rotenone, as well as in the substantia nigra dopaminergic neurons of PD patients, while NEDD4-1 seems to be downregulated [283]. An in vitro study showed that RTP801 can be subjected to lysosomal degradation and is conjugated with K63-linked polyubiquitin chains by NEDD4-1 [85]. It has been proposed that RTP801 is stress-induced upregulated at the early stage of PD to maintain cellular function, but sustained elevation and mTOR and AKT inhibition might lead to dopaminergic cell death [284]. However, it has also been suggested that NEDD4-1 acts as a downstream target of the PI3K/PTEN–mTORC1 signalling pathway to promote neurite growth and regeneration [182].

As ion channel dysfunction become increasingly intertwined with PD pathology, it is worth investigating the capacity of NEDD4 ligases to interact with them, particularly in vivo. Dopaminergic neurons are characterised electrophysioligically by their spontaneous discharge from pacemaker activity to burst-firing [285,286,287,288,289,290]. Tonic action potential firing which contributes to the pacemaker activity of dopaminergic neurons is controlled by ion channels [291]. Ion channels can play critical roles in the neuronal excitability, cell volume and the regulation of neurotransmitter release.

Voltage-gated sodium channels (Na_v_s) reside in the cell membrane are essential for the creation and propagation of action potentials. Studies examining the interaction of NEDD4-2 with Na_v_s have been carried out in vivo, in vitro and ex-vivo and demonstrated that Na_v_s 1.2, 1.3, 1.5, 1.6, 1.7, 1.8 are substrates of NEDD4-2 mediated through interaction with PPSY, PLSY and PGSP motifs [122,123,124]. In a patch-clamp experiment, Na_v_1.2 was shown to be downregulated when NEDD4-2 was expressed in HEK293 cells [122]. Na_v_1.6 is essential for neuronal excitability and a number of motor disorders are associated with mutations in Na_v_1.6. Homozygous-null mutations in Na_v_1.6 lead to juvenile mortality in mice between P19 and P21 [292]. Phosphorylated Pro-Gly-Ser^553^-Pro motif on Na_v_1.6 is a putative binding site for NEDD4 ubiquitin ligases. One study hypothesised that NEDD4-2 contributes to the ubiquitination and subsequent internalisation of Na_v_1.6. In cultured cells, NEDD4-2 was indeed shown to interact with Na_v_1.6 through a C-terminal Pro-Ser-Tyr^1945^ motif, causing a reduction in Na_v_1.6 current density. This regulation appears to require both the Pro-Gly-Ser-Pro motif in L1 and the Pro-Ser-Tyr motif in the C terminus of Na_v_1.6. When NEDD4-2 binding to the Pro-Ser-Tyr motif was prevented, a stress-mediated increase in Na_v_1.6 current density was observed. Phosphorylation of the Pro-Gly-Ser-Pro motif in the L1 of Na_v_1.6 seems necessary for stress-induced current modulation. Positive or negative regulation appears to depend on the availability of the PRO-Ser-Tyr motif in the C-terminus to bind NEDD4-2 [124].

A study examined cognitive impairments in a rat model of PD [293]. In these 6-OHDA lesioned rats, Na_v_1.1 was substantially elevated in reactive hippocampal astrocytes 28 days after lesioning, which reduced after 49 days. No changes were observed in Na_v_1.6 levels at 28 days, but was elevated in hippocampal neurons at a later time-point of 49 days post-lesion. The predominantly embryonically expressed Na_v_1.3 appeared to be re-expressed in hippocampal CA neurons at 49 days post-lesion. In this study 6-OHDA lesioned rats were treated with the Na_v_ blocker, phenytoin. These rats exhibited improved spatial learning and memory in the Morris water maze compared to lesioned rats not given the phenytoin [293].

Na_v_s, in particular Na_v_1.6, appear to play an important role in neuronal physiology and may play a role in the genesis of congnitive defecits in PD. It would be of interest to examine the in vivo role of Na_v_s in dopaminergic neurons specifically and how this might be altered in the presence or absence of NEDD4 ligases. It would also be of interest to investigate how these channels behave in α-synuclein-mediated models of PD and how this could impact the fate of dopaminergic neurons and their interaction with proximal glia.

In addition to voltage-gated sodium channels, NEDD4 ligases have also been demonstrated to interact with potassium ion channels. In the brain KCNQ2 and KCNQ4 potassium ion channels are mostly limited to the substantia nigra and ventral tegmental area of the midbrain [294,295,296,297]. In the striatum, dopaminergic nerve termini express KCNQ2 and KCNQ3 [297]. The muscarine-sensitive K(+) current (M-current) stabilises neuronal resting potential, therefore limiting neuronal excitability. M-current is mediated through heteromeric ion channels comprised of KCNQ3 subunits which associate with either KCNQ2 or KCNQ5 subunits. In a study examining the regulation of KCNQ2/3/5 channels it was revealed that NEDD4-2 but not NEDD4-1 could reduce K(+) currents mediated by KCNQ2/3 and KCNQ3/5 in a *Xenopus* oocyte expression system. Through deletion experiments it was shown that the KCNQ3 subunit is required for NEDD4-2 to regulate the heteromeric channels. Co-immunoprecipitation and Glutathione S-transferase fusion pulldown experiments demonstrated that NEDD4-2 and KCNQ2/3 interact directly. NEDD4-2 was also able to ubiquitinate KCNQ2/3 in transfected HEK293T cells [119]. Other Potassium channels in this family have also been shown to interact with NEDD4 ligases (particularly NEDD4-2) [116,120,121], however these channels have not currently been implicated in PD or dopaminergic system pathology. It appears however that in the nervous system NEDD4-2 is potentially an important M-current activity regulator.

## 6. Conclusions and Future Directions

The collected data support the notion that further research is required to clarify the unique and common features of NEDD4-1 and NEDD4-2 especially in the midbrain dopaminergic system affected in PD. The large number of possible mechanisms for regulating NEDD4 ubiquitination activity, substrate specificity, and protein interactions make an *in silico* prediction of possible outcomes extremely difficult. The regulation, function, and substrate specificity of NEDD4-1 and NEDD4-2 need to be studied in vivo in a tissue- and cell-type-specific fashion before strategies can be designed to propose them as therapeutic targets for neurodegenerative diseases such as PD. The possible substrates suggest that NEDD4-1 and/or NEDD4-2 could be beneficial or harmful in the disease context. Currently, it is not clear if NEDD4-1 and/or NEDD4-2 protein levels should be increased or decreased to improve the conditions in dopaminergic neurons under pathophysiological conditions such as PD. Research on NEDD4 proteins in neurodegenerative diseases remains an exciting field in which many surprising findings can still be expected. Unbiased approaches should therefore be applied to remain open to all possible outcomes. In the recent years, the toolkit for studying NEDD4-1 and NEDD4-2 has dramatically improved, with the availability of conditional animal models for NEDD4-1 and NEDD4-2, good antibodies, and specific knockdown possibilities, which will facilitate further investigations.

## Figures and Tables

**Figure 1 genes-13-00513-f001:**
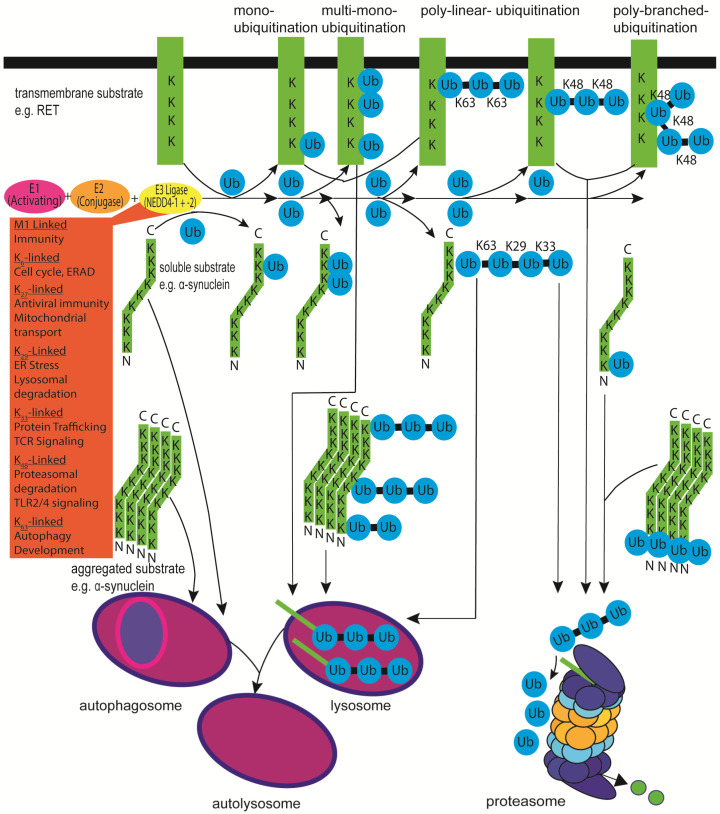
NEDD4-mediated ubiquitination of protein substrates. Attachment of ubiquitin (Ub) to proteins is usually catalysed by an enzymatic cascade of a ubiquitin-activating enzyme E1, a ubiquitin-binding/conjugating enzyme E2, and a ubiquitin–protein ligase enzyme E3 that catalyses the transfer of the C-terminal carboxyl group of ubiquitin to the lysine (K) ε-amino group of the specific substrate. The process of ubiquitination can occur on transmembrane proteins (e.g., RET, ion channels) and on intracellular proteins (e.g., α-synuclein). The fate of the protein is dependent upon the number of ubiquitin moieties attached to each other on a substrate and which amino acid in ubiquitin the chain is extended: one of the seven lysines (K6, K11, K27, K29, K33, K48, K63) or, through its N-terminal, methionine (M1). Monoubiquitination and multimonoubiquitination of a transmembrane protein generally result in its transport, internalisation, and recycling. Linear and branched polyubiquitination with K48-linked chains results in proteasomal degradation of the substrate, and that with K63 extension regulates protein–protein interactions, protein activity, DNA repair, autophagy, immunity, inflammation, and protein trafficking to the lysosome [9]. The primary role(s) of each of the eight distinct polyubiquitin chains formed at one of the seven lysine residues or the primary methionine are indicated (orange box) [4,5,8]. ER = endoplasmic reticulum; ERAD = Endoplasmic-reticulum-associated protein degradation; TCR = T-cell receptor; TLR2/4 = Toll-like receptor 2 and 4.

**Figure 2 genes-13-00513-f002:**
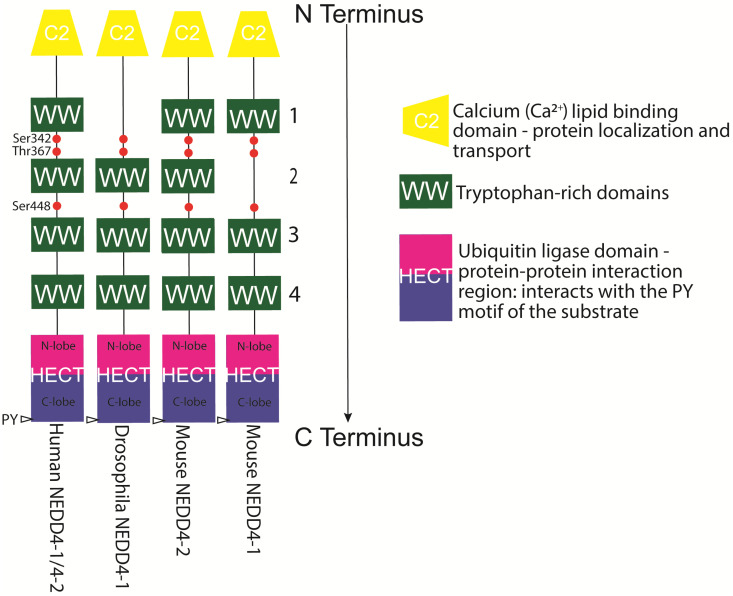
Schematic structural representation of NEDD4-1 and NEDD4-2 proteins in humans, mice, and fruit flies. The NEDD4 family of ligases is defined by its modular structure, a lipid-binding/Ca^2+^ (C2) domain at the N-terminus, a number of WW domains in the middle section, and a HECT ubiquitin ligase at the C-terminus, the latter of which is required for its E3 ubiquitin ligase function. NEDD4’s WW domains can interact with PY (proline-tyrosine) motifs to recruit them for ubiquitination. This includes NEDD4’s own PY motifs located on the C terminus. Alternative splicing in mice has led some NEDD4-2 variants to lack a C2 domain, although in neurons, NEDD4-2 predominantly contains a C2 domain. WW domains regulate substrate recruitment for ubiquitination and may be expanded in higher-order organisms [111,215]. Common NEDD4 phosphorylation sites are indicated in red.

**Table 1 genes-13-00513-t001:** Substrates, adaptors, and modifiers/regulators of NEDD4-1.

Name	Substrate	Adaptor	Modifier/Regulator	Binding Motifs	Modification	Function	Experimental Model Used	References
ABCB1 (ATP-binding cassette sub-family B member 1/P-Glycoprotein)	🗸			PDY	Polyubiquitination	An ATP Binding Cassette transporter that exports β-amyloid from Blood-brain barrier endothelial cells. Potential for intervention in Alzheimer’s disease	In vitro: Sf21 cells	[31]
α-synuclein	🗸			PDNEAYEMP, PLY, PPLP, PPSP, PFY	Monoubiquitination and Polyubiquitination (K63–linked)	Lysosomal degradation. Potential protection mechanism against Parkinson’s Disease pathogenesis	In vitro: SH–SY5Y cells In vivo: *Drosophila* and Rat	[28,29,30]
Beclin 1	🗸			LPLY	Beclin 1: polyubiquitination (K63 and K11–linked)	Subunit of the class III phosphatidylinositol 3-kinase complex. Autophagy-related protein. Proteasomal degradation. Inhibited autophagy and cell survival	In vitro: HeLa cells	[32]
β2-AR (β2-Arrestin Receptor)	🗸				Ubiquitination	Involved in internalised receptor degradation and lysosomal trafficking. Degraded via the lysosome	In vitro: HEK293 cells	[33]
β-arrestin 1		🗸				Adaptor protein for NHE1 ubiquitination	In vitro: HEK293 cells, mouse embryonic fibroblasts	[34]
β-arrestin 2		🗸	🗸			Adaptor protein for β2-adrenergic receptor ubiquitination	In vitro: HEK293 cells	[33]
Caspase-1, -3, -4, -6, -7, -11	🗸		🗸		Truncation	NEDD4-1 can be cleaved by caspases and can K48-polyubiquitinate caspase 11	In vitro: Jurkat cells A549	[14,35,36,37]
Cbl-b (Casitas B-lineage Lymphoma b)	🗸		🗸		Polyubiquitination	Impedes NEDD4-1 interaction with PTEN and also polyubiquitinates Cbl-b for degradation	In vivo: *Cbl-b^C373A^* and *NEDD4^Gt(IRES^*^β*geo)249Lex*^ mice	[38,39]
CNrasGEF (Cyclic Nucleotide rat sarcoma virus Guanine nucleotide Exchange Factor)	🗸			PPGY, PPDY	Polyubiquitination	RAS guanine nucleotide exchange factors that are degraded via the proteasome. NEDD4-1 overexpression promoted migration and invasion of glioma cells	In vitro: HEK293T and Glioma cells	[40,41]
Connexin43	🗸			PPGY	Ubiquitination	Gap junction protein. Proteasome and lysosome degradation	In vitro: WB-F344 rat liver epithelial cells	[42]
c-Src (Proto-oncogene tyrosine-protein kinase Src)			🗸			Tyrosine kinase that activates NEDD4-1 through phosphorylation of its HECT and C2 domains. Phosphorylation inhibits auto-regulation	In vitro: HeLa, HEK293T and Platinum E cells	[43]
Δ Np63 transcriptional target	🗸		🗸	PPPY	Ubiquitination and polyubiquitination	A homologue of p53 tumour suppressor. Protein destabilisation of ∆Np63α and degradation. Downregulates NEDD4-1 leading to the suppression of nuclear PTEN in basal layer keratinocytes	In vitro: HEK293-EBNA, HaCaT, A431 andH1299 cells In vivo: Zebrafish embryos	[44,45]
EPS15 (Epithelial growth factor receptor substrate 15)	🗸				Monoubiquitination and polyubiquitination	An endocytic protein that is targeted for degradation by NEDD4-1	In vitro: HeLa and B82L cells	[46]
FGFR1 (Fibroblast Growth Factor Receptor 1)	🗸		🗸	VLLVRPSRLSSSG	Ubiquitination	FGFR1 is a tyrosine kinase involved in cell proliferation and differentiation during development. Inhibited neural stem cell differentiation. Activates c-Src that subsequently activates NEDD4-1	In vitro: Human embryonic stem cells In vivo: Zebrafish embryos	[47,48]
GAG (group specific antigen) protein	🗸			PPPY, PTAP	Monoubiquitination	From the HTLV-1 (human T-lymphotropic virus type 1) involved in hijacking mutivesicular body (MVB) pathway proteins required for viral budding	In vitro: HEK293T cells	[49]
γ2-adaptin	🗸	🗸		PPAY	Monoubiquitination and multi-polyubiquitination	A member of the clathrin adaptor protein family. Forms a complex with NEDD4-1 and is involved in endosomal/multivesicular body (MVB) pathway and the assembly and release of the HBV	In vitro: HuH-7 and HeLa cells	[50,51]
HBV X protein (Hepatitis B virus)	🗸				Ubiquitination	A multifunctional regulator that is encoded by the HBV genome. It is degraded via the K48 proteasomal pathway	In vitro: HEK293T, HBV-related HCC cell lines - HepG2.215, HepG3B, SNU182, SNU387, PLC/PRF/5, and MHCC97H	[52]
HER3 (human epidermal growth factor receptor 3)	🗸			PPRY	Polyubiquitination	A member of the EGRF family. Degraded via the proteasome. Inhibited cancer cell proliferation and tumour growth. NEDD4-1 knockdown induces apoptosis in DU145 cells	In vivo: Chinese Hamster ovary cells In vitro: MDA-MB-453, MCF-7, and DU145 cells	[53,54]
HGS (Hepatocyte growth factor-regulated tyrosine kinase substrate)	🗸			PPEY	Ubiquitination	Binding partner for NEDD4-1 involved in EGFR lysosome degradation	In vitro: Chinese hamster ovary and HeLa cells	[55]
IFITM3 (Interferon (IFN)-induced transmembrane protein 3)	🗸			PPNY	Polyubiquitination	A cell-intrinsic factor that limits influenza virus and other viral infections such as SARS	In vitro: HEK293T, A549, NCI-H358, NCI-H2009 cells and MEFs	[56]
IGF1R (Insulin-like growth factor 1 receptor)	🗸				Ubiquitination	A tyrosine kinase receptor. Expression can be downregulated by NEDD4-1 through the indirect effect on the oxidisation of very-low-density lipoproteins. Ubiquitination and degradation require a C1060 site	In vitro: Hepatocytes from Landes goose embryos In vivo: Intracerebral haemorrhage mice, Sprague Dawley and Tg2576 mice	[57]
IGPR-1 (Immunoglobulin and Proline-rich receptor-1, also known as TMIGD2/CD28H)	🗸			PPR	Polyubiquitination	A cell adhesion molecule involved in, for example, autophagy, angiogenesis and cell adhesion. Is degraded via the lysosomal pathway	In vitro: HEK293 cells	[58]
IRS-2 (insulin receptor substrate)	🗸				Monoubiquitination	NEDD4-1 recruits IRS-2 to the membrane to enhance IGF signalling	In vivo: Zebrafish embryos In vitro: HEK293 cells	[59]
ISG15 (Interferon-stimulated gene 15)			🗸		ISGylation	Can attach to NEDD4-1 inhibiting its ubiquitination properties	In vitro: HEK293, HeLa cells and MEFs	[60,61,62]
KLF8 (Krueppel-like factor 8)	🗸				Ubiquitination	The function of this transcription factor is regulated by NEDD4-1	In vitro: HEK293 cells	[63,64]
LATS1 (large tumour suppressor kinase 1)	🗸				Ubiquitination	A serine/threonine kinase involved in the suppression of tumours	In vitro: HEK293 cells	[65]
LC3 (Microtubule-associated protein 1A/1B-light chain 3)			🗸	WEII, WVVL, WFFL, WDKL		An autophagy-related protein. LC3 binds to NEDD4-1, but is not a ubiquitination substrate of NEDD4-1	In vitro: HEK293 cells	[66]
LDLRAD4 (Low density lipoprotein receptor class A domain containing 4)	🗸				Ubiquitination	Degraded via the lysosome and is a negative regulator of TGF-β signalling	In vivo: Female BALB/c nude mice In vitro: L02 and HepG2 cells	[67]
LMP2A (Latent membrane protein 2A)	🗸			PPPPY	Ubiquitination	A latent Epstein–Barr virus (EBV) infection protein involved in B cell signal transduction	In vitro: BJAB, Ramos, Raji, Jurkat, HPB.ALL and M12 cells	[68]
MDM2 (Mouse double minute 2 homolog)	🗸				Polyubiquitination (K63-linked)	Is an E3 ubiquitin ligase involved in negative regulation of p53. MDM2 is stabilised via NEDD4-1 interacting with its RING domain. NEDD4-1 overexpression reduces p53 levels	In vivo: NEDD4-1 KO mouse embryonic fibroblasts	[69]
MEKK5 (mitogen-activated protein kinase 5) (Apoptosis Signal-regulating Kinase 1 (ASK1)			🗸			A serine/threonine kinase that regulates NEDD4-1 cell migration signalling in lung cancer	In vitro: HEK293T, NCI-H1650, and A549 cells	[58]
N4BP (NEDD4-binding protein)	🗸			PPLP, PPEY, PPPY	Monouniquitination and Polyubiquitination	N4BP is degraded via the proteasome. NEDD4-1 regulates N4BP1 at promyelocytic leukaemia nuclear bodies	In vitro: HEK293 cells and MEFs	[70,71]
NAB (N-aryl benzdimidazole)		🗸?				NAB2 reduces the ratio of K63-linked ubiquitin chains on A53T α-synuclein by an unknown mechanism. NAB2 binds NEDD4-1 but does not alter conformation or enzymatic activity.	In vitro: SH-SH5Y cells	[72,73]
NEDD4-1 (Neural precursor cell-expressed developmentally-down-regulated protein 4-1)	🗸		🗸		Autoubiquitination	K29-linked autoubiquitination. C2 and HECT domains bind resulting in autoubiquitination	In vitro: HEK293T, HeLa, THP-1 and A549 cells	[74,75]
NHE1 (Sodium-Hydrogen antiporter 1)	🗸				Multi-monoubiquitination and polyubiquitination	The Na(+)/H(+) exchanger 1 is ubiquitinated for degradation by NEDD4-1 but requires β-arrestin 1	In vitro: HEK293 cells	[34]
N-Myc (N-myc proto-oncogene protein/ basic helix-loop-helix protein 37)	🗸				Polyubiquitination	An oncoprotein that is degraded via the proteasome. NEDD4-1 suppresses neuroblastoma and pancreatic cancer cell proliferation	In vitro: Neuroblastoma BE(2)-C, CHP134, pancreatic cancer MiaPaca-2 and HEK293 cells	[76]
Notch	🗸			PPSY	Polyubiquitination	A plasma membrane receptor that is ubiquitinated for degradation via the proteasome	In vivo: *Drosophila* and Conditional NEDD4-1 overexpression in Wistar Rat	[77,78,79]
Rap2a (RAS-related protein 2a)	🗸				Monoubiquitination of K63	Rap2a Is a member of the RAS-related protein family. NEDD4-1 Inhibits GTP-Rap2a activity subsequently promoting the migration and invasiveness of glioma cells	In vitro: Human glioma cell lines U251 and U87	[80]
RAS (Rat sarcoma virus)	🗸			PPGY, PPDY	Polyubiquitination and monoubiquitination	Small guanosine triphosphatases involved in a multitude of different cellular processes by acting as a molecular switch. RAS is regulated via NEDD4-1ubiquitination sending it for degradation to the lysosome. This regulation suppressed tumorigenesis	In vitro: HEK293T, HeLa, NIH 3T3, MEF and HepG2 cells	[40,81]
RET (Rearranged during transfection)	🗸				Polyubiquitination	A receptor tyrosine kinase. The short form (Ret9) becomes localised and internalised into the endosomal network through clathrin-coated pits following NEDD4-1 ubiquitination. This causes inhibition of Ret9-mediated neurotrophic signalling at the cell surface and promotion of post-internalisation signalling. This mechanism could potentially impact neurotrophic signalling of dopaminergic neurons and play a role in Parkinson’s disease	In vitro: HEK293 and SH-SY5Y cells	[82]
RNAPII (Ribonucleic acid Polymerase II)	🗸				Monouniquitination and polyubiquitination	A multiprotein involved in the transcription of DNA into mRNA that is degraded via the proteasome after being ubiquitinated by NEDD4-1. This ubiquitination is dependent on NEDD4-1 interacting with the ElonginA/B/C-Cullin 5 complex	In vitro: HEK293, MRC5 and *S. cerevisiae*	[83,84]
RTP801/REDD1	🗸				Polyubiquitination	A pro-apoptotic protein that is targeted for degradation by NEDD4-1via K63 ubiquitin linkages. NEDD4-1 loss may elevate RTP801 proteins leading to an increase in neuronal death in Parkinson’s disease	In vivo: *NEDD4-1*^f/f^, *Emx1*Cre mice In vitro: PC12, HEK293 cells and rat primary cortical neurons	[85]
SAG (S-Arrestin)	🗸				Polyubiquitination	An anti-apoptotic cellular survival protein that is degraded by the proteasome. NEDD4-1 reduction of SAG resulted in etoposide-induced apoptosis in cancer cells. SAG does not bind to WW domains as it lacks PY motifs but interacts with NEDD4-1 via its RING domain	In vitro: HEK293T	[86]
SCAMP3 (Secretory Carrier Membrane Protein 3)	🗸			PPAY, PSAP, PTEP	Multi-monoubiquitination	Integral membrane proteins involved in the cell surface recycling system. SCAMP3 is a NEDD4-1 substrate that is involved in the degradation of EGFR via the lysosome	In vitro: HeLa and HEK293T cells	[87]
Spy1A	🗸				Polyubiquitination	A cyclin-like protein that is needed for a cell to progress through the G_1_/S phase. Spy1A is required for p53-mediated tumour suppression. Spy1A is degraded in a cell cycle-dependent manner during mitosis via the ubiquitin-proteasome system	In vitro: Human mammary breast cancer, MCF7, and HEK293cells	[88,89]
SQSTM1 (p62)	🗸				Polyubiquitination (K63-type)	An autophagy-related protein. NEDD4-1 ubiquitinates its PB1 domain. Lack of NEDD4-1 leads to accumulation of aberrant enlarged inclusion bodies	In vitro: HEK293T, HEK293A and A549 cells	[66,90]
VEGF-R2 (vascular endothelial growth factor receptor-2)	🗸				Monoubiquitination	This receptor is degraded by NEDD4-1 but Grb10 regulates this process by interacting with NEDD4-1	In vitro: HEK-293 EBNA cells	[91]

?, experimental uncertainty. Further investigation required to confirm result. Abbreviations: PDY, Proline Aspartate Tyrosine; PDNEAYEMP, Proline Aspartate Asparagine Glutamate Alanine Tyrosine, Glutamate, Methionine Proline; PLY, Proline Leucine Tyrosine; PPLP, Proline Proline Leucine Proline; PPSP, Proline Proline Serine Proline; PFY, Proline Phenylalanine Tyrosine; LPLY, Leucine Proline Leucine Tyrosine; PPGY, Proline Proline Glycine Tyrosine; PPDY, Proline Proline Aspartate Tyrosine; PPPY, Proline Proline Proline Tyrosine; VLLVRPSRLSSSG, Valine Leucine Leucine Valine Arginine Proline Serine Arginine Leucine Serine Serine Serine Glycine; PTAP, Proline Threonine Alanine Proline; PPAY, Proline Proline Alanine Tyrosine; PPRY, Proline Proline Aspartate Tyrosine; PPEY, Proline Proline Glutamate Tyrosine; PPNY, Proline Proline Asparagine Tyrosine; PPR, polyproline rich; WEII, Tryptophan Glutamate Isoleucine Isoleucine; WVVL, Tryptophan Valine Valine Leucine; WFFL, Tryptophan Phenylalanine; WDKL, Tryptophan Aspartate Lysine Leucine; PPPPY, Proline Proline Proline Proline Tyrosine; PPSY, Proline Proline Serine Tyrosine; PSAP, Proline Serine Alanine Proline; PTEP, Proline Threonine Glutamate Proline.

**Table 2 genes-13-00513-t002:** Substrates, adaptors, and modifiers/regulators of NEDD4-2.

Name	Substrate	Adaptor	Modifier/Regulator	Binding Motifs	Modification	Function	Experimental Model Used	References
14-3-3		🗸	🗸			14-3-3 is an inhibitory binding partner for NEDD4-2 through a PI3-kinase/SGK1-dependent manner. This interaction is dependent on the phosphorylation of key residues Ser^342^ and Ser^448^ on NEDD4-2	In vitro: Kidney tubule epithelial mpkCCDc_14_, HECT293 and *E. coli* BL21(DE3) cells	[92,93]
α-Arrestins	🗸	🗸		PPLP, PPEY, PPLY, PPSY, PPNY, PPPY	Ubiquitination; adaptor/regulator	α-Arrestins are scaffolding molecules involved in regulating receptor trafficking and cell signalling. These can be both substrates and adaptors for NEDD4-2. α-arrestins are implicated in the regulation of DMT1. It is activated through polymerisation or membrane tethering and is ubiquitinated by NEDD4-2	In vitro: CHO, Caco-2, HepG2 and HEK293T cells In vivo: *Arrdc1^tm1(KOMP)Vlcg^* (VG17312) and *Arrdc4^tm1(KOMP)Vlcg^* (VG18749) embryonic stem cells	[27,94,95]
AMPK (AMP-activated protein kinase)		🗸				AMPK is a metabolic sensor that inhibits ENaC. It activates NEDD4-2 by phosphorylation, promotes ENaC-NEDD4-2 interaction and subsequent ENaC degradation. AMPK also signals in the mTOR pathway where it plays a role in cell death.	In vitro: HEK293 cells In vivo: *Xenopus* oocytes	[96]
ATA2 (Analogue Terminal Adaptor II)	🗸				Polyubiquitination	NEDD4-2 regulates amino acid transporter ATA2 activity on the cell surface by proteasomal degradation	In vitro: 3T3-L1 adipocytes Ex vivo: *Xenopus* oocytes	[97]
CFTR (Cystic fibrosis transmembrane conductance regulator)	🗸				Ubiquitination	CFTR is downregulated by NEDD4-2 possibly via the proteasome and lysosome degradation (contested by [98]). Ref. [99], however, show that CFTR is degraded via NEDD4-2 but has to interact with 14-3-3ε and be activated by SGK1 phosphorylation	In vitro: CFPAC-1 (ΔF508) and CFBE41o-ΔF cells	[98,99,100]
CHT1 high-affinity choline transporter 1)	🗸				Ubiquitination	NEDD4-2-mediated ubiquitination regulates cell surface expression of CHT1 thereby impeding choline uptake and HC-3 binding	In vitro: HEK293 Cells	[101]
CLC-5 (Chloride Voltage-Gated Channel 5)	🗸			PPLPPY	Ubiquitination	Voltage-gated channels that function as dimers. NEDD4-2 decreases cell surface expression of ClC-5 through ubiquitination	Ex vivo: *Xenopus* oocytes In vivo: NEDD4–2 null mice	[102,103,104]
CLC-K (Chloride channel protein ClC-Ka) /barttin	🗸			PPYVRL (located on barttin)	Ubiquitination	A chloride channel that requires barttin to be functional. Downregulation of ClC-Ka/barttin comes as a result of NEDD4-2-mediated ubiquitination	Ex vivo: *Xenopus* oocytes	[105,106]
CRTC3 (CREB Regulated Transcription Coactivator 3)	🗸			PPPY	Polyubiquitination	NEDD4-2 is responsible for the downregulation of CRTC3 in a proteasome–dependant manner in response to prolonged cAMP signalling	In vitro: HEK293T cells	[107]
DAT (Dopamine Transporter)	🗸				Polyubiquitination	A transporter for dopamine. PKC-dependent DAT ubiquitination by NEDD4-2 requires its WW3 and WW4 domains. May have implications in Parkinson’s disease	In vitro: HEK 293 and PAE cells	[108]
DMT1 (Divalent metal transporter 1)	🗸				Polyubiquitination	A metal transporter that is ubiquitinated by NEDD4-2 but requires the adaptor protein Ndfip1	In vitro: SH-SY5Y and HEK293T cells	[109]
DVL2 (Dishevelled 2)	🗸			PPPY	Polyubiquitination	NEDD4-2 negatively regulates Wnt signalling by targeting dishevelled for proteasomal degradation. Wnt5a induces JNK-mediated phosphorylation of NEDD4-2, which in turn promotes Dvl2 degradation	In vitro: HEK293T and HeLa cells	[110,111]
EAAT1/2 (Excitatory amino acid transporter 1 and 2. Also known as Glutamate transporter 1 and 2)	🗸			PPPD	Ubiquitination	Regulation of EAAT1/2 through NEDD4-2 depends on SGK kinases. NEDD4-2 knockdown with shRNA decreases GLT-1 ubiquitination, promoting glutamate uptake and increases GLT-1 expression. This may play a role in glutamatergic signalling in dementia	Ex vivo: *Xenopus* oocytes In vitro: MPP^+^ treated astrocytes	[112,113,114]
GluA1 (Glutamate receptor 1)	🗸			PKY	Ubiquitination	NEDD4-2 ubiquitinates GluA1 at lysine-868 and mediates its surface expression. This may play a role in glutamatergic signalling in dementia	In vitro: HEK293 cells In vivo: *NEDD4-2^andi^* mice	[115]
hERG(1) (human ether-à-gogo-related gene (1))	🗸			PPAY	Monoubiquitination and polyubiquitination	The human ether-a-go-go-related gene protein (hERG) is a voltage-gated cardiac potassium channel. Caveolin-3 (Cav3), hERG, and NEDD4-2 form a complex. hERG expression in the plasma membrane is regulated by Cav3 through NEDD4-2 ubiquitination	In vitro: HEK293 cells (Patch clamp) In vivo: Guinea pig	[38,116]
IKKβ (inhibitor of nuclear factor kappa-B kinase subunit β)		🗸	🗸			IKKβ activates NEDD4-2 via phosphorylation that results in the regulation of ENaC	In vitro: HEK-293, HEK-293T and mpkCCD_c14_ cells. Ex vivo: *Xenopus* oocytes (TEV)	[117]
JNK1 (Janus Kinase 1)		🗸	🗸			JNK1 activates NEDD4-2 via phosphorylation that results in the regulation of ENaC	In vitro: HEK293 cells and mpkCCD_c14_ cells. Ex vivo: *Xenopus* oocytes (TEV)	[118]
KCNQ (Voltage-Gated Potassium Channel Subunits) 1, 2/3, 3/5	🗸			PPDPPY	Polyubiquitination	Amplitude of K^+^ currents mediated by KCNQ2/3 and KCNQ3/5 were reduced by NEDD4-2. NEDD4-2 is activated by AMPK leading to reduced KCNQ1 expression	In vitro: HEK293 cells Ex vivo: *Xenopus* oocytes	[116,119,120,121]
Na_v_s (Voltage-gated sodium channels) 1.2, 1.3, 1.5, 1.6, 1.7, 1.8	🗸			PP*S*Y, LP*S*Y PGSP	Ubiquitination	Are vital in creating and propagating action potentials and reside in the membrane. NEDD4-2 interaction inhibits activity of multiple Na_v_s, including the cardiac (Na_v_1.5) and neuronal Na_v_s (Na_v_1.2, Na_v_1.7, and Na_v_1.8)	In vitro: HEK293 cells In vivo: SNS-*NEDD4-2^–/–^* mice, Pulldown of mouse brain lysates Ex vivo: *Xenopus* oocytes	[122,123,124]
NCC (Sodium Chloride symporter)	🗸				Ubiquitination	NCC ubiquitination at the cell surface Is achieved by NEDD4-2 and its deficiency upregulate NCC. NEDD4-2 may require another protein to achieve this	In vitro: HEK293 and mDCT_15_ cells Ex vivo: *Xenopus* Oocyte In vivo: *Pax8-rtTA* and *TRE-LC1* transgenic mice	[125,126]
NEDD4-2(Neural precursor cell-expressed developmentally-downregulated gene/protein 4-2)	🗸		🗸	LPPY	Inhibitory self-ubiquitination of NEDD4–2	Promotes NEDD4-2 stabilisation through auto-ubiquitination involving its own PY motif located on its HECT domain. This interaction may be between an active and non-active form	In vitro: HEK293 cells Ex vivo: *Xenopus* oocytes	[104,127,128]
NEDD8 (Neural precursor cell-expressed developmentally-downregulated gene/protein 8)			🗸		Neddylation	Neddylation is a process whereby the ubiquitin-like protein, Nedd8, is conjugated to NEDD4-2 resulting in its degradation	In vitro: Mouse M1 kidney and mouse NCTC1469 liver cells In vivo: *CYP4F2* transgenic mice	[129]
NHE3 (Sodium–hydrogen antiporter 3)	🗸			PPNY	Ubiquitination	An Na^+^/H^+^ exchanger that is expressed within the kidney where it is involved in blood pressure regulation through NaCl and HCO_3_^−^ absorption. NHE is ubiquitinated by NEDD4-2. Disruption of NEDD4-2 interaction elevates human NHE3 expression and activity	In vitro: HEK 293 and PS120 cells	[130]
NKCC1/2 (Na-K-Cl cotransporter ½)	🗸					An Na^+^/K^+^/2Cl^−^ co-transporter where NEDD4-2 is involved in its downregulation. NEDD4-2 indirectly suppresses NKCC1 expression	In vitro: HEK293T In vivo: Tam-induced *NEDD4-2^f/f^*; Vil-Cre^ERT2^ mice	[36]
OAT (Organic ion transporters) 1/3	🗸				Polyubiquitination and multiubiquitination	NEDD4-2 regulates cell surface OAT1/3 expression and its transport activities	In vitro: COS-7 and HEK293T cells	[131,132]
Occludin	🗸			PPPY	Polyubiquitination	An integral membrane protein that NEDD4–2 ubiquitinates. NEDD4-2 overexpression reduced occluding at tight junctions	In vitro: HEK293 and mplCCD_c14_ cells	[133]
PKA (Protein Kinase A)			🗸			Inhibitory phosphorylation of NEDD4-2. cAMP regulates ENaC through phosphorylation & inhibition of NEDD4-2	In vitro: COS-7, FRT epithelial cells	[134]
SCAMP3 (Secretory Carrier Membrane Protein 3)	🗸			PPAY	Multi-monoubiquitination	SCAMP3 ubiquitination is involved in the degradation of EGFR via the lysosome	In vitro HeLa and HEK293T cells	[87]
SGK1 (Serum/Glucocorticoid Regulated Kinase 1)		🗸	🗸	PPFY		Regulates the activity of several ion transport proteins. Inhibitory Phosphorylation of NEDD4-2 causes its interaction with 14-3-3 and subsequent degradation	In vitro: Kidney tubule epithelial and COS7 cells	[92,135,136]
SGLT1 (sodium-glucose linked transporter 1)	🗸				Ubiquitination	NEDD4-2 ubiquitinates and downregulates SGLT1	Ex vivo: *Xenopus* oocytes	[137]
Smad (Mothers against decapentaplegic homolog) 2, 3, 4, 7	🗸			PPPY	Polyubiquitination	NEDD4–2, interacts with Smads, inducing their polyubiquitination and degradation. This is not the case for Smad3	In vitro: COS7 (Smad2) and HEK293T (Smad 2,3,4&7) cells	[136,138,139,140,141]
SP-C (Surfactant Protein C)	🗸			PPDY	Monoubiquitination or biubiquitination at K6	NEDD4-2–mediated ubiquitination regulates luminal relocation of SP-C, leading to its processing and secretion	In vitro: HEK293 cells	[142]
TGF-βR1 (Transforming Growth Factor-β Receptor 1)	🗸					Plays a role in epithelial-mesenchymal transition via phosphorylation of small mothers against decapentaplegic (SMAD). NEDD4-2 suppresses its signalling	In vivo: Sprague Dawley rats. In vitro: NRK-52E cells	[143]
TrkA(Tropomyosin-related kinase Trk A)	🗸			PPVY, PPSY, PPIY	Multi-monoubiquitination	Tropomyosin-related kinase (Trk) A is a receptor specifically for nerve growth factor and is downregulated by NEDD4-2	In vitro: PC12-615 cells and primary cortical neurons	[144,145,146]
Tweety	🗸			PPTY	Ubiquitination	A family of chloride ion channels. NEDD4-2-mediated ubiquitination of TTYH2 regulates both cell surface and total levels of Tweety proteins	In vitro: HEK293 cells	[147]
ULK1 (Unc-51 like autophagy activating kinase 1)	🗸				Polyubiquitination	A serine–threonine kinase involved in autophagy. NEDD4-2 ubiquitinates ULK1 and targets it for proteasomal degradation	In vitro: HeLa cells	[148]
Usp2-45 (Ubiquitin-specific protease 2-45)		🗸				Adaptor of NEDD4-2 for ENaC Ubiquitination	In vitro: HEK293 cells	[149]
WNK1 (With No Lysine Kinase)	🗸			PPQY, PFY	Ubiquitination	Serine-threonine kinases that regulate potassium, sodium, and blood-pressure homeostasis. Hormonal (Aldosterone, insulin and vasopressin) regulation of NEDD4-2 and WNK to regulate NCC (thiazide-sensitive NaCl cotransporter)	In vitro: mpkCCD_c14_ and HEK294T Cells In vivo: *NEDD4-2^fl/fl^* Pax8-*rtTATRE-LC1* (Renal-specific NEDD4-2 KO) mice	[150]

Abbreviations: PPLP (Proline Proline Leucine Proline), PPEY (Proline Proline Glutamate Tyrosine), PPLY (Proline Proline Leucine Tyrosine), PPSY (Proline Proline Serine Tyrosine), PPNY (Proline Proline Asparagine Tyrosine), PPPY (Proline Proline Proline Tyrosine), PPLPPY (Proline Proline Leucine Proline Proline Tyrosine), PPYVRL (Proline Proline Tyrosine Valine Arginine Leucine), PPPD (Proline Proline Proline Aspartate), PKY (Proline Lysine Tyrosine), PPAY (Proline Proline Alanine Tyrosine), PPDPPY (Proline Proline Aspartate Proline Proline Tyrosine), LPTY (Leucine Proline Threonine Tyrosine), LP*S*Y (Leucine Proline Serine Tyrosine), PGSP (Proline Glysine Serine Proline), LPPY (Leucine Proline Proline Tyrosine), PPFY (Proline Proline Phenylalanine Tyrosine) (Proline Proline Asparagine Tyrosine), PPDY (Proline Proline Aspartate Tyrosine), PPVY (Proline Proline Valine Tyrosine), PPIY (Proline Proline Isoleucine Tyrosine), PPQY (Proline Proline Glutamine Tyrosine), PFY.

**Table 3 genes-13-00513-t003:** Substrates, adaptors, and modifiers/regulators of both NEDD4-1 and NEDD4-2.

Name	Substrate	Adaptor	Modifier/Regulator	Binding Motifs	Modification	Function	Experimental Model Used	References
ACK-1 (activated Cdc42-associated kinase 1) (NEDD4-1)	🗸			PPAY	Monoubiquitination, Polyubiquitination	ACK-1 is a cytoplasmic tyrosine kinase and is a NEDD4-1 and NEDD4-2 substrate. Its degradation through the proteasome results in downregulation of ACK-1. These authors suggest only NEDD4-1 to ubiquitinate ACK-1	In vitro: COS7, HEK293T, HeLa, T47D, and A549 cells	[151]
ACK-1 (activated Cdc42-associated kinase 1) (NEDD4-2)	🗸			PPAY	Polyubiquitination	Its degradation through the proteasome results in downregulation of ACK-1 although this is contested [151]	In vitro: HeLa and COS7 cells	[152]
α-synuclein filaments (NEDD4-1)	🗸			PDNEAYEMP, PLY, PPLP PPSP, PFY	Monoubiquitination and Polyubiquitination (K63-linked)	Degraded via the lysosome. Potential protection mechanism against Parkinson’s Disease pathogenesis	In vitro: SH-SY5Y In vivo: Drosophila and Rat	[28,29,30]
α-synuclein filaments (NEDD4-2)	🗸				Polyubiquitination		In vitro: SH-SY5Y cells	[29]
AKT (Protein Kinase B) (NEDD4-1)	🗸			LPEY, LPFY	Polyubiquitination (K63 and K48-linked) and multi-monoubiquitination	A critical effector kinase that regulates numerous cellular processes such as cell growth, death, differentiation, and migration.NEDD4-1 regulates nuclear trafficking of the activated form of AKT via the proteasome & enhances bortezomib sensitivity	In vitro: MCF-7, HeLa, and NEDD4^+/+^, and NEDD4^−/−^ cells and MEFs	[153,154]
AKT (Protein Kinase B) (NEDD4-2)			🗸			Inhibitory phosphorylation of NEDD4-2. Inhibits NEDD4-2, increasing ENaC expression and Na^+^ absorption	In vitro: FRT cells	[155]
AMPAR (NEDD4-1)	🗸				Ubiquitination	An ionotropic glutamate receptor. Ubiquitination leads to AMPAR internalisation and subsequent degradation	In vivo: Rat dissociated hippocampal or cortical neurons In vitro: HEK293 cells	[156,157,158]
AMPAR (NEDD4-2)	🗸				Ubiquitination	Ubiquitinates the GRIA1 subunit of AMPAR thereby mediating neuronal excitation	In vivo: Sprague Dawley (SD) rats, Nedd4-2^andi^ and GluA1 knockout mice In vitro: HEK293 cells	[115,159,160]
AQP2 (Aquaporin 2) (NEDD4-1)	🗸				Polyubiquitination	Ubiquitinates and degrades AQP2 but requires NDFIP1 and NDFIP2 adaptors	In vitro: HEK293 and mpkCCD cells	[161]
AQP2 (Aquaporin 2) (NEDD4-2)	🗸				Polyubiquitination	Ubiquitinates and degrades AQP2 but requires NDFIP1 and NDFIP2 adaptors	In vitro: HEK293 and mpkCCD cells	[161]
Ca^2+^ (Calcium ions) (NEDD4-1)			🗸			Binds to the C2 domain of NEDD4-1 leading to the activation of its ligase activity and inhibition of auto-regulation. PIP2/IP3 ratio dictates its function in either the proximity of the membrane (PIP2) or cytoplasm (IP3)	Used biophysical techniques to complement the literature (NMR Spectroscopy)	[128]
DLG3 Discs Large MAGUK Scaffold Protein 3) (NEDD4-1 and NEDD4-2)	🗸			PPGY, PPDY	Monoubiquitination	NEDD4-1 interaction results in Dlg3 monoubiquitination, apical membrane recruitment, and tight junction consolidation	In vitro: MDCK cells In vivo: Co-IP of mouse brain lysates	[162]
DVL2 (NEDD4-1 and NEDD4-2)	🗸				Ubiquitination	A protein involved in the Wnt pathway. Reported to be ubiquitinated for degradation via NEDD4-1 and NEDD4-2	In vivo: *NEDD4-1^fl^**^/fl^* and *NEDD4-2^fl^**^/fl^* mice In vitro: HEK293T, DLD1 and HCT116 cells	[163]
ENaC (Epithelial Sodium Channel) (NEDD4-1)	🗸			PPNY, PPRY, PPAY	Ubiquitination of α and γ subunits	The epithelial Na^+^ channel is downregulated through proteosome degradation. Lack of binding motif in Liddle’s syndrome causes hyperactivity	In vitro: rat foetal distal lung epithelial and *Xenopus*-derived A6 cells Ex vivo: *Xenopus* oocyte	[164,165]
ENaC (Epithelial Sodium Channel) (NEDD4-2)	🗸			PPAY, PPNY, PPRY, PPKY	Monoubiquitination	NEDD4-2 catalyses ubiquitination and reduces expression of ENaC at the cell surface through lysosomal degradation. NEDD4-2 also targets Na^+^/Cl^−^ cotransporter (NCC) in the kidney	In vitro: HEK293T cells	[116,126,166,167]
Grb10 (Growth Factor Receptor Bound Protein 10) (NEDD4-1)		🗸				An adaptor for NEDD4-1 to ubiquitinate and degrade IGF-IR. Required for indirect ubiquitination and degradation of VEGFR2	In vitro: p6, p6/Grb10, HEK-293 EBNA and HUVEC cells	[42,168,169,170]
Grb10 (Growth Factor Receptor Bound Protein 10) (NEDD4-2)		🗸	🗸	PQTPF		Associates with NEDD4-2 leading to its regulation and inhibition of ubiquitination and degradation of K_v_1.3 channels	In vitro: HEK293 cells	[171]
LGR5 (Leucine-rich repeat containing G protein-coupled receptor 5) (NEDD4-1 and NEDD4-2)	🗸				Ubiquitination	A receptor for R-spondin and is a protein involved in the Wnt pathway. It is apparently degraded via the lysosome and proteasome involving NEDD4-1 and NEDD4-2	In vivo: *NEDD4-1^fl/fl^* and *NEDD4-2l^fl/fl^* mice In vitro: HEK293T, DLD1 and HCT116 cells	[163]
MTMR4 (NEDD4-1)	🗸			PPLY	Ubiquitination	Myotubularin-related protein 4, an inositol phosphatase that regulates endosomal signalling	In vitro: HEK293 or HeLa cells	[172,173]
MTMR4 (NEDD4-2)			🗸	PPLY		MTMR4 reduces NEDD4-2-mediated proteasome degradation of wild type and mutant KCNQ1 and hERG channels via dephosphorylation	In vitro: Induced pluripotent stem cardiomyocytes from human dermal fibroblasts	[174]
Na_V_ (Voltage-gated Sodium channels)1.2 and 1.7 (NEDD4-1)	🗸			PP*S*Y (Proline Proline Serine Tyrosine)	Ubiquitination	Regulates sodium channels through lysosomal degradation	Ex vivo: *Xenopus Oocytes*	[175]
Na_v_s (Voltage-gated Sodium channels) 1.2, 1.3,1.5, 1.6, 1.7, 1.8 (NEDD4-2)	🗸			PP*S*Y, LPSY	Ubiquitination	NEDD4-2 ubiquitination inhibits activity of multiple Na_v_s, including the cardiac (Na_v_1.5) and neuronal Na_v_s (Na_v_1.2, Na_v_1.7, and Na_v_1.8)	In vitro: HEK-293 cells. In vivo: SNS-*NEDD4-2^–/–^* mice, Pulldown of mouse brain lysates Ex vivo: *Xenopus* oocytes	[42,116,122,123,124]
Ndfip1/2 (NEDD4 family-interacting proteins) (NEDD4-1)		🗸	🗸			NEDD4 family–interacting protein that activates the HECT domain of NEDD4-1. Ndfip1 binds substrates such as PTEN to act as an adaptor for NEDD4-1 ligase activity	In vitro: HEK-293T, SH-SY5Y, PC3 cells and primary MEFs In vivo: *C57BL/6J,* Emx1-Cre (C57BL/6) and *Nestin-Cre* mice (B6.Cg-Tg(Nes-cre)1Kln/J)	[176]
Ndfip1/2 (NEDD4 family-interacting proteins) (NEDD4-2		🗸	🗸	PPPY, PPSY, LPTY, PSY, PTY		Ndfip activates HECT domain of NEDD4-2	Ex vivo: *Xenopus* oocytes	[42,161,176]
OAT1 (organic anion transporter 1) (NEDD4-1)	🗸				Ubiquitination	Organic anion transporter (OAT). NEDD4-1 is an important regulator for hOAT1 ubiquitination, expression, and function via its WW2 and WW3 domains	In vitro: COS-7 and HEK293T cells	[131]
OAT1/3 (Organic ion transporters 1/3) (NEDD4-2)	🗸				Ubiquitination	NEDD4-2 ubiquitination regulates cell surface OAT1/3 expression with their WW3 and WW4 domains	In vitro: COS-7 and HEK293T cells	[131,132]
PTEN (Phosphatase and tensin homolog) (NEDD4-1& NEDD4-2)	🗸?	🗸		PRR	Multi-monoubiquitinationPolyubiquitination(Ubiquitination independent of NEDD4-1)	PTEN is a tumour suppressor. Nuclear importation occurs via monoubiquitinationNEDD4-1 single and NEDD4-1 & NEDD4-2 double knock out mice do not exhibit stability, subcellular activity or localisation differences of PTEN	In vitro: PC3 and HEK293T cells In vivo: *Xenopus* tectumIn vivo: Murine	[111,176,177,178,179,180,181,182]

?, experimental uncertainty. Further investigation required to confirm result. Abbreviations: PPAY (Proline Proline Alanine Tyrosine), PDNEAYEMP (Proline Aspartate Asparagine Glutamate Alanine Tyrosine Glutamate Methionine Proline), PLY (Proline Leucine Tyrosine), PPLP (Proline Proline Leucine Proline), PPSP (Proline Proline Serine Proline), LPEY (Leucine Proline Glutamate Tyrosine), LPFY (Leucine Proline Phenylalanine Tyrosine), PPGY (Proline Proline Glycine Tyrosine), PPDY (Proline Proline Aspartate Tyrosine), PPNY (Proline Proline Asparagine Tyrosine), PPRY (Proline Proline Arginine Tyrosine), PPAY (Proline Proline Alanine Tyrosine), PPKY (Proline Proline Lysine Tyrosine), PQTPF (Proline Glutamine Threonine Proline Phenylalanine), PPLY (Proline Proline Leucine Tyrosine), PP*S*Y (Proline Proline Serine Tyrosine), LP*S*Y (Leucine Proline Serine Tyrosine), LPTY (Leucine Proline Threonine Tyrosine), PTY (Proline Threonine Tyrosine), PRR (Proline rich region).

## Data Availability

Not applicable.

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
