# Peer review of "The Role of NEDD4 E3 Ubiquitin–Protein Ligases in Parkinson’s Disease"

_genes, 2022, doi:10.3390/genes13030513_

Round 1

Reviewer 1 Report

This is a very nice, comprehensive and complete review on NEDD4 ubiquitin protein ligases. I do not have any corrections to suggest.

Reviewer 2 Report

Conway et al., wrote a comprehensive review on the role ubiquitin ligases on the degradation of alpha-synuclein. The authors first summarize the diverse field of ubiquitin ligases on alpha-synuclein before focusing on the NEDD family, in particular NEDD4. This review is aptly-timed and includes updated literature. The authors do provide extensive summarized literature on NEDD4 family members in other contexts, such as regulation of SMAD4 signaling and IGF-1, which can be reduced if space is needed.  Otherwise, the review is well-written.

Reviewer 3 Report

The present manuscript is an outstanding review on the role and regulation of NEDD4 ligases, focused on dopaminergic neurons and Parkinson disease. 

The structure of the work is really good, the analysis of the  bibliography is profound and sound. Overall, the contribution is important.

Nonetheless, I would suggest to edit figures and tables to higher quality formats (they should have the excellent quality of the text). There are nice free softwares (some of them available online) to generate figures that could help to create nicer representation of proteins, sub-cellular compartments, membranes, pathways, etc. In tables, I would suggest to choose a smaller size, and maybe different typos, and reorganise columns in order to avoid the loss of information. These two points are minor, and they could be solved in the editorial process, without any problem.